

# Adjoint subordination to calculate backward travel time probability of pollutants in water with various velocity resolutions

Yong Zhang[1], Graham E. Fogg[2], Hongguang Sun[3], Donald M. Reeves[4], Roseanna M. Neupauer[5], Wei Wei[6]

[1]Department of Geological Sciences, University of Alabama, Tuscaloosa, AL 35487, USA
[2]Hydrologic Sciences, University of California, Davis, CA 95616, USA
[3]College of Mechanics and Materials, Hohai University, Nanjing 210098, China
[4]Department of Geological and Environmental Sciences, Western Michigan University, Kalamazoo, MI 49008, USA
[5]Department of Civil, Environmental, and Architectural Engineering, University of Colorado Boulder, Boulder, CO 80309,
USA
[6]School of Environment, Nanjing Normal University, Nanjing 210023, China

*Correspondence to*: Yong Zhang (yzhang264@ua.edu) and HongGuang Sun (shg@hhu.edu.cn)

**Abstract.** Backward probabilities such as backward travel time probability density function for pollutants in natural
aquifers/rivers had been used by hydrologists for decades in water-quality related applications. Reliable calculation of
backward probabilities, however, has been challenged by non-Fickian pollutant transport dynamics and variability in the
resolution of velocity at study sites. To address these two issues, we built an adjoint model by deriving a backward-in-time
fractional-derivative transport equation subordinated to regional flow, developed a Lagrangian solver, and applied the
model/solver to backtrack pollutant transport in various flow systems. The adjoint model applies subordination to a reversed
regional flow field, converts forward-in-time boundaries to either absorbing or reflective boundaries, and reverses the tempered
stable density to define backward mechanical dispersion. The corresponding Lagrangian solver is computationally efficient in
projecting backward super-diffusive mechanical dispersion along streamlines. Field applications demonstrate that the adjoint
subordination model can successfully recover release history, dated groundwater age, and spatial location(s) of pollutant
source(s) for flow systems with either upscaled constant velocity, non-uniform divergent flow field, or fine-resolution
velocities in a non-stationary, regional-scale aquifer, where non-Fickian transport significantly affects pollutant dynamics and
backward probability characteristics. Caution is needed when identifying the phase-sensitive (aqueous versus absorbed)
pollutant source in natural media. Possible extensions of the adjoint subordination model are also discussed and tested for
quantifying backward probabilities of pollutants in more complex media, such as discrete fracture networks.

## 1 Introduction

Backward probabilities of pollutants in natural aquifers/rivers, such as backward travel time probability density function
(BTTP), has been used by hydrologists for decades in water quality related applications. For example, BTTP defines the
possible length of time for contaminants to reach a sampling location (e.g., a monitoring well screen or stream sampling



location) from their source location(s) (Neupauer and Wilson, 2001; Ponprasit et al., 2023). It contains useful information on contaminant fate and transport which can help water management, remediation, and assessment. For instance, a common application of BTTP is to recover contamination history and identify responsible parties, where the BTTP's peak captures the

most likely release time of contaminants from the source (Skaggs and Kabala, 1994; Woodbury and Ulrych, 1996; Woodbury et al., 1998; Sun et al., 2006a, 2006b; Jha and Datta, 2015; Yeh et al., 2015; Jamshidi et al., 2020; Chen et al., 2023). BTTP can also be used to date groundwater since BTTP characterizes the age distribution of groundwater due to borehole mixture and/or hydrodynamic dispersion in regional-scale aquifers (Weissmann et al., 2002; Cornaton and Perrochet, 2006; LaBolle et al., 2006; Zinn and Konikow, 2007a, 2007b; Janssen et al., 2008; McMahon et al., 2008; Maxwell et al., 2016; Ponprasit et

al., 2022; Mao et al., 2023). In addition, BTTP provides a more comprehensive method to assess aquifer vulnerability than classical statistics-based approaches through the generation of three-dimensional (3-$d$), transient vulnerability maps for groundwater to non-point source contamination (Fogg et al., 1999; Zhang et al., 2018). BTTP can also be used to estimate solute concentration trends (Green et al., 2014), and rates of oxygen and nitrate reduction in regional groundwater settings (Green et al., 2016). These applications demonstrate that mathematical models can reliably be applied to quantify backward

probabilities including BTTP, and most importantly, a general BTTP model is still needed due to the challenges mentioned below, motivating this study.

There are two main challenges in numerical quantification of backward probabilities including BTTP for contaminant transport in surface water and groundwater. First, a novel model is needed to account for the impact of complex transport dynamics of contaminants on BTTP. Previous BTTP models, which are usually the inverse or backward advection-dispersion

equations (ADEs), assumed Fickian diffusion of contaminants (meaning that the plume variance grows linearly in time); see the extensive review by Moghaddam et al. (2021). Real-world contaminant transport, however, is usually non-Fickian at almost all relevant scales, where the temporal evolution of the plume variance can be either slower than linear (which is called "sub-diffusion") or faster than linear ("super-diffusion"), as recently reviewed by Guo et al. (2021). Particularly, super-diffusion can be driven by turbulence or flooding events in streams (Phillips et al., 2013; Boano et al., 2014), preferential flow pathways

consisting of fractures in fractured porous media (Reeves et al., 2008), or high-permeability paleochannels within alluvial deposits (Bianchi et al., 2016). Sub-diffusion is more common in natural water systems due to pervasive solute retention or storage mechanisms such as physical/chemical sorption-desorption, heterogeneous advection (meaning a broad range of advective velocities), and/or multi-rate mass exchange between mobile and relative immobile flow zones (Haggerty et al, 2000; Zhou et al., 2021). Fickian-diffusion based classical models cannot capture super/sub-diffusive non-Fickian transport if the

velocity field is not sufficiently resolved (e.g., coarser than the centimetre scale; see Zheng et al. (2011)) or if the model underestimates the spatial interconnectivity of high-permeability deposits (Yin et al., 2020). To address this issue, various nonlocal transport models, which are typically non-Markovian models considering the spatiotemporal memory during solute transport, have been developed to efficiently simulate forward-in-time non-Fickian transport (Neuman and Tartakovsky, 2009), but their corresponding BTTP models remained obscure (Zhang et al., 2022; Zhang, 2022).





The second challenge is how to incorporate the observed velocity field, the resolution of which typically varies significantly between field sites, into the backward probabilities (including BTTP) calculation. Many field sites have only limited hydrologic information, requiring a fully upscaled BTTP model which can function using a coarsely resolved velocity field or even a uniform velocity. Contrarily, some well-studied field sites may have abundant geologic/hydrologic data, providing a detailed spatiotemporal distribution of velocities that should be incorporated into the numerical model to improve

the reliability of BTTP calculations. Ideally, an efficient BTTP model should be able to incorporate the velocity field without any resolution constraints.

To fill these two knowledge gaps, this study proposes an adjoint subordination approach by deriving a backward-in-time model (which is also called "adjoint") of the 3-$d$, time fractional-derivative equation (FDE) subordinated to water flow with or without a highly resolved velocity field. Such a forward-in-time FDE was proposed by Zhang et al. (2015) as a general

forward model for pollutant transport in various geological media. Notably, two other vector nonlocal transport models can also incorporate the local variation of velocity in non-Fickian diffusion, which are the well-known continuous-time random walk (CTRW) framework (Hansen and Berkowitz, 2020) and the multi-scaling FDE model (Zhang, 2022). The CTRW framework allows various memory functions to define solute transition times, but does not separate sub-diffusion (due to solute retention) and super-diffusion (due to for example preferential flow paths) (Lu et al., 2018). The subordinated time-FDE, as

shown in **section 1**, is selected for this study because (i) it can capture both sub-diffusion (using the time fractional derivative) and sub-grid super-diffusion (via subordination), and (ii) it is computationally more efficient than the multi-scaling FDE (introduced in **section 4**).

The rest of this work is organized as follows. **Section 2** applies a sensitivity analysis approach to build the adjoint of the subordinated time-FDE, and then develops and validates a Lagrangian solver of the resultant BTTP model. **Section 3** checks

the feasibility of the adjoint model and its solver by quantifying BTTP, identifying the release history of contaminants in an alluvial aquifer and a river with a uniform velocity, and calculating groundwater ages dated by environmental tracers in a regional-scale alluvial aquifer with a fine resolution of velocities. **Section 4** discusses the identification of contaminant source locations based on the backward location probability density function (BLP) and the backward probability model extension. **Section 5** draws the main conclusions.

**2 Methodology development**

This section derives the model and solver for backward-in-time subordination to water flow in heterogeneous media. The concept of subordination to regional flow was first proposed by Baeumer et al. (2001) and then extended to multi-dimensional flow by Zhang et al. (2015). Subordination is a statistical method that can randomize the operational time experienced by each individual particle in a random process (Feller, 1971). When applying subordination to regional flow, fast displacement of

pollutant particles along streamlines is captured during the randomized operational time, as shown and explained in the following model (1a).





### 2.1 Forward and backward models

#### 2.1.1 Three-dimensional transport and adjoint models

We propose the following 3-*d* subordinated time-FDE to track pollutants in streams and aquifers with a vector velocity, after adding source/sink and reaction terms and initial/boundary conditions in the vector model proposed by Zhang et al. (2015):

$$b\frac{\partial(\theta C)}{\partial t} + \beta\frac{\partial^{\gamma,\lambda}(\theta C)}{\partial t^{\gamma,\lambda}} = -\nabla_{\vec{V}}(\theta C) + \sigma^*\left(\nabla_{\vec{V}}\right)^{\alpha,\kappa}(\theta C) + q_I C_I - q_o C - \theta r C \tag{1a}$$

$$C(\vec{x}, t = 0) = \frac{M_0}{\theta}\,\delta(\vec{x} - \vec{x}_0) \tag{1b}$$

$$C(\vec{x}, t)\big|_{\xi_1} = g_1(t) \tag{1c}$$

$$\left[\sigma^*\left(\nabla_{\vec{V}}\right)^{\alpha-1,\kappa}(\theta C)\right]\cdot n_2\Big|_{\xi_2} = g_2(t) \tag{1d}$$

$$\left[V(\vec{x}, t)\,\theta C - \sigma^*\left(\nabla_{\vec{V}}\right)^{\alpha-1,\kappa}(\theta C)\right]\cdot n_3\Big|_{\xi_3} = g_3(t) \tag{1e}$$

where $C$ [$ML^{-3}$] denotes the solute concentration, $b$ (= 0 or 1) [dimensionless] is a factor controlling the type of the time FDE, $\theta$ [dimensionless] is the effective porosity, $\beta$ [$T^{\gamma-1}$] is the fractional capacity coefficient, $\sigma^*$ [$L$] is a scaling factor for subordination, $\vec{V}$ [$LT^{-1}$] is the velocity vector, $\nabla_{\vec{V}}$ is an advection operator defined via $\nabla_{\vec{V}} = \nabla(\vec{V}C)$, $q_I$ [$T^{-1}$] is the source

inflow rate, $C_I$ is the inflow concentration, $q_O$ is the sink outflow rate, $r$ [$T^{-1}$] is the first-order decay constant, $M_0$ is the initial source mass, $g_i$ ($i = 1, 2, 3$) is a known function at the type-$i$ boundary (to define the constant concentration or pollutant flux at the boundary), $\xi_i$ ($i = 1, 2, 3$) is the domain of the type-$i$ boundary, $\vec{x}$ [$L$] denotes the spatial coordinate, $t$ [$T$] is the (forward) time, and $n_2$ and $n_3$ are the outward unit normal vectors on the type-2 and type-3 boundaries, respectively. We name Eq. (1) the subordinated fractional-dispersion equation (S-FDE).

The S-FDE (1a) captures the concurrent sub-diffusion and super-diffusion (driven by different mechanisms) using different terms. The symbol $\frac{\partial^{\gamma,\lambda}}{\partial t^{\gamma,\lambda}}$ in Eq. (1a), which is the mixed Caputo fractional derivative with an index $\gamma$ [dimensionless] ($0 < \gamma \leq 1$) and a temporal truncation parameter $\lambda$ [$T^{-1}$] (Baeumer et al., 2018), defines sub-diffusion due to solute retention. The operator $\left(\nabla_{\vec{V}}\right)^{\alpha,\kappa}$, which denotes subordination to the flow field with an index $\alpha$ [dimensionless] ($1 < \alpha \leq 2$) for the tempered stable density (with the maximumly positive skewness $\beta^* = +1$) and a spatial truncation parameter $\kappa$ [$L^{-1}$], describes

fast displacements to downstream motion. The method of subordination to regional flow expands the standard symmetric mechanical dispersion to non-symmetric, super-diffusive mechanical dispersion along streamlines caused by the local variation of velocities (such as super-diffusion along preferential flow paths). Notably, the molecular diffusion term can be added to Eq. (1) to define the full range of hydrodynamic dispersion, if the molecular diffusive strength is not negligible.

To derive the backward model for the S-FDE (1) using the adjoint approach (Neupauer and Wilson, 2001), we first

convert it to the model governing the state sensitivity $\phi = \frac{\partial c}{\partial f}$, where $f$ is a system parameter and selected as the initial mass





$M_0$ as in Neupauer and Wilson (2001) and Zhang (2022). This can be done by taking the first-order derivative of each term in the S-FDE (1) with respect to $M_0$, which leads to:

$$\left( b\frac{\partial}{\partial t} + \beta\frac{\partial^{\gamma,\lambda}}{\partial t^{\gamma,\lambda}} \right)(\theta\phi) = -\nabla_{\bar{V}}(\theta\phi) + \sigma^*(\nabla_{\bar{V}})^{\alpha,\kappa}(\theta\phi) - (q_o + \theta r)\phi \tag{2a}$$

$$\phi(\vec{x}, t = 0) = \frac{\partial C(\vec{x}_i)}{\partial M_0} = \frac{1}{\theta}\,\delta(\vec{x} - \vec{x}_0) \tag{2b}$$

$$\phi(\vec{x}, t)|_{\xi_1} = 0 \tag{2c}$$

$$\left[\sigma^*(\nabla_{\bar{V}})^{\alpha-1,\kappa}(\theta\phi)\right]\cdot n_2\Big|_{\xi_2} = 0 \tag{2d}$$

$$\left[V\theta\phi - \sigma^*(\nabla_{\bar{V}})^{\alpha-1,\kappa}(\theta\phi)\right]\cdot n_3\Big|_{\xi_3} = 0 \tag{2e}$$

where the time fractional derivative operator commutes.

We then add the adjoint state of the concentration in the S-FDE (2a) by taking the inner product of each term of Eq. (2a) with an arbitrary function $A$, which represents the adjoint state:

$$\int_0^T\int_\Omega \left[ Ab\frac{\partial(\theta\phi)}{\partial t} + A\beta\frac{\partial^{\gamma,\lambda}}{\partial t^{\gamma,\lambda}}(\theta\phi) + A\nabla_{\bar{V}}(\theta\phi) - A\sigma^*(\nabla_{\bar{V}})^{\alpha,\kappa}(\theta\phi) + A(q_o + \theta r)\phi \right] d\Omega\, dt = 0 \tag{3}$$

where $\Omega$ denotes the whole model domain. Next, we change the position of the state sensitivity $\phi$ and the adjoint sate $A$ in the first four terms of Eq. (3). For example, the 1st term in Eq. (3), denoted as $I_1$, can be re-arranged using integration by parts:

$$I_1 = \int_\Omega \left[\int_0^T Ab\frac{\partial(\theta\phi)}{\partial t}dt\right]d\Omega = \int_\Omega \left\{[Ab\theta\phi]|_{t=0}^{t=T} - \int_0^T \theta\phi b\frac{\partial A}{\partial t}dt\right\}d\Omega\,. \tag{4}$$

The 2nd term in Eq. (3) contains the time fractional derivative and can be re-arranged using the fractional-order integration by parts, as shown in Zhang (2022):

$$I_2 = \int_\Omega \left[\int_0^T A\beta\frac{\partial^{\gamma,\lambda}(\theta\phi)}{\partial t^{\gamma,\lambda}}dt\right]d\Omega = \int_\Omega \left\{A|_{t=T}\,\beta\, I_+^{1-\gamma,\lambda}(\theta\phi)|_{t=T} - [\theta\phi]|_{t=0}\,\beta\, I_-^{1-\gamma,\lambda}(A)|_{t=0} + \int_0^T \theta\phi\beta\frac{\partial^{\gamma,\lambda}A}{\partial(-t)^{\gamma,\lambda}}dt\right\}d\Omega, \tag{5}$$

where the symbol $I_+^{1-\gamma,\lambda}(f)$ denotes the positive fractional integral of order $1-\gamma$: $I_+^{1-\gamma,\lambda}(f) = e^{-\lambda T}\int_0^T f e^{\lambda t}\frac{(T-t)^{-\gamma}}{\Gamma(1-\gamma)}dt$, the symbol $I_-^{1-\gamma,\lambda}(f) = e^{\lambda T}\int_0^T f e^{-\lambda t}\frac{t^{-\gamma}}{\Gamma(1-\gamma)}dt$ denotes the negative fractional integral of order $1-\gamma$, and $\Gamma(\cdot)$ is the gamma function.

The 3rd term in Eq. (3), which describes the net advective flux, can be re-arranged using the integer-order integration by parts:

$$I_3 = \int_0^T \left\{\int_\Omega \nabla\cdot[A\theta V\phi]\,d\Omega - \int_\Omega \theta V\phi\,\nabla A\,d\Omega\right\}dt = \int_0^T \left\{\oint_\xi [A\theta V\phi]\cdot n\,d\xi - \int_\Omega \theta V\phi\,\nabla A\,d\Omega\right\}dt, \tag{6}$$

where the Gauss' divergence theorem is used: $\int_\Omega \nabla\cdot f\,d\Omega = \oint_\xi f\cdot n\,d\xi$, and $n$ is the outward normal direction on the boundary $\xi$. Eqs. (4)~(6) are the same as those shown in Zhang (2022), which is expected since the same time fractional derivative term was used in these FDEs.

The 4th term in Eq. (3) contains the subordination operator and can be re-arranged using the integration by parts for twice, as shown in Zhang (2022):



$$I_4 = \int_0^T \left[ \int_\Omega A\sigma^*(\nabla_{\bar{V}})^{\alpha,\kappa}(\theta\phi)\,d\Omega \right] dt = \int_0^T \left\{ \oint_\xi \sigma^*\left[ A\,I_+^{2-\alpha,\kappa}\left( \nabla_{\bar{V}}(\theta\phi) \right) \right]\cdot n\,d\xi + \oint_\xi \sigma^*\left[ \nabla_{\bar{V}}(\theta\phi)\,I_-^{2-\alpha,\kappa}(A) \right]\cdot n\,d\xi \right\} dt$$

$$+ \int_0^T \left\{ \oint_\xi \sigma^*\left[ \theta\phi\,(\nabla_{\bar{V}})^{\overline{\alpha-1},k}(A) \right]\cdot n\,d\xi \right\} dt - \int_0^T \left\{ \int_\Omega \sigma^*\theta\phi\,(\nabla_{\bar{V}})^{\bar{\alpha},k}(A)\,d\Omega \right\} dt \;. \tag{7}$$

Here the operator $(\nabla_{\bar{V}})^{\bar{\alpha},k}$ denotes subordination to the reversed flow field $(\bar{V})$ where the tempered stable density (with order $\alpha$) has the maximumly negative skewness $\beta^* = -1$, meaning that fast displacements are from downstream to upstream (for backward tracking).

Neupauer and Wilson (2001) showed that the adjoint state $A$ is a measure of the change in concentration for a unit change in source mass $M_0$. In sensitivity analysis, the marginal sensitivity of a performance measure $A$ with respect to $M_0$ is (Neupauer and Wilson, 2001):

$$\frac{dP}{dM_0} = \int_0^T \int_\Omega \left[ \frac{\partial h(M_0, C)}{\partial C}\,\phi \right] d\Omega\,dt\;, \tag{8}$$

where $h(M_0, C)$ is a functional of the state of the system. Inserting $I_1 \sim I_4$ expressed by Eqs. (4)~(7) into the inner product equation (3), and then subtracting this updated Eq. (3) from the marginal sensitivity equation (8), we obtain:

$$\frac{dP}{dM_0} = \int_\Omega \int_0^T \phi \left\{ \frac{\partial h}{\partial C} + b\theta\frac{\partial A}{\partial t} - \beta\theta\frac{\partial^{\gamma,\lambda}A}{\partial(-t)^{\gamma,\lambda}} + \theta V\nabla A - \sigma^*\theta(\nabla_{\bar{V}})^{\bar{\alpha},k}(A) - (q_o + \theta r)A \right\} d\Omega\,dt$$

$$- \int_\Omega \left\{ [Ab\theta\phi]|_{t=T} - [Ab\theta]|_{t=0}\frac{\partial C_i}{\partial M_0} + A|_{t=T}\beta I_+^{1-\gamma,\lambda}(\theta\phi)|_{t=T} - [\theta\phi]|_{t=0}\beta I_-^{1-\gamma,\lambda}(A)|_{t=0} \right\} d\Omega$$

$$- \int_0^T \oint_\xi \left[ A\theta V\phi - A\,I_+^{2-\alpha,\kappa}\left( \nabla_{\bar{V}}(\theta\phi) \right) - \nabla_{\bar{V}}(\theta\phi)\,I_-^{2-\alpha,\kappa}(A) - \theta\phi\,(\nabla_{\bar{V}})^{\overline{\alpha-1},k}(A) \right]\cdot n\,d\xi dt. \tag{9}$$

To eliminate $\phi$ from Eq. (9), we define $A$ such that the terms containing $\phi$ vanish. Since the double integral in Eq. (9) (shown by the first line in Eq. (9)) can be eliminated when the summation of all the terms inside the bracket is zero, this

produces the adjoint equation of the S-FDE (1a):

$$b\theta\frac{\partial A}{\partial t} - \beta\theta\frac{\partial^{\gamma,\lambda}A}{\partial(-t)^{\gamma,\lambda}} = -\theta V\nabla A + \sigma^*\theta(\nabla_{\bar{V}})^{\bar{\alpha},k}(A) + (q_o + \theta r)A - \frac{\partial h}{\partial C}\;. \tag{10}$$

Assuming (i) the backward time $s = T - t$ where $T$ is the detection time, (ii) steady-state groundwater flow (so that $\theta V\nabla A - q_o A = \nabla(\theta VA) - q_I A$), and (iii) un-compressible aquifer skeleton (so that $\partial\theta/\partial t = 0$), we can re-write Eq. (10) as:

$$b\frac{\partial(\theta A)}{\partial s} + \beta\frac{\partial^{\gamma,\lambda}(\theta A)}{\partial s^{\gamma,\lambda}} = \nabla_{\bar{V}}(\theta A) - \theta\sigma^*(\nabla_{\bar{V}})^{\bar{\alpha},\kappa}A - (q_I + \theta r)A + \frac{\partial h}{\partial C} \tag{11a}$$

$$A(\vec{x}, s)|_{s=0} = 0 \tag{11b}$$

$$A(\vec{x}, s)|_{\xi_1} = 0 \tag{11c}$$

$$\left[ -A\theta V + \sigma^*\theta\,(\nabla_{\bar{V}})^{\overline{\alpha-1},\kappa}(A) \right]\cdot n_2\Big|_{\xi_2} = 0 \tag{11d}$$

$$\left[ \sigma^*\theta\,(\nabla_{\bar{V}})^{\overline{\alpha-1},\kappa}(A) \right]\cdot n_3\Big|_{\xi_3} = 0 \tag{11e}$$

where the initial condition (11b) $A(\vec{x}, t)|_{t=T} = A(\vec{x}, s)|_{s=0} = 0$ and the boundary conditions (11c)~(11e) are obtained by

making sure that the remaining terms in Eq. (9) defines the following marginal sensitivity:

$$\frac{dP}{dM_0} = \int_\Omega \left\{ [(Ab\theta)|_{t=0} + \theta|_{t=0}\,\beta I_-^{1-\gamma,\lambda}(A)|_{t=0}]\frac{\partial C_i}{\partial M_0} \right\} d\Omega\;. \tag{12}$$



Therefore, to convert the forward-in-time S-FDE (1) to its backward counterpart (11), we need to (i) reverse the flow field, (ii) convert the source/sink terms and boundary conditions, and (iii) reverse the skewness in the stable density defining backward mechanical dispersive jumps. The first two changes were identified before by Neupauer and Wilson (2001) for the

classical ADE (although the exact forward-backward transition is new here), and the last change is new. In the following we name the backward-in-time model (11) as the adjoint S-FDE.

### 2.1.1 One-dimensional simplifications

The 1-$d$ simplification of the vector forward-in-time S-FDE (1) takes the form:

$$b\frac{\partial(\theta C)}{\partial t} + \beta\frac{\partial^{\gamma,\lambda}(\theta C)}{\partial t^{\gamma,\lambda}} = -\frac{\partial(V\theta C)}{\partial x} + \sigma^*\left(\frac{\partial}{\partial x}\right)_V^{\alpha,\kappa}(\theta C) + q_I C_I - q_o C - \theta r C$$

$$C(x, t = 0) = \frac{M_0}{\theta}\,\delta(x - x_0)$$

$$C(x,t)|_{\xi_1} = g_1(t)$$

$$\left[\sigma^*\left(\frac{\partial}{\partial x}\right)_V^{\alpha-1,\kappa}(\theta C)\right]\Bigg|_{\xi_2} = g_2(t)$$

$$\left[V\theta C - \sigma^*\left(\frac{\partial}{\partial x}\right)_V^{\alpha-1,\kappa}(\theta C)\right]\Bigg|_{\xi_3} = g_3(t)$$

If the velocity $V$ in the equations listed above is constant, this 1-$d$ S-FDE reduces to the following 1-$d$ standard FDE:

$$b\frac{\partial(\theta C)}{\partial t} + \beta\frac{\partial^{\gamma,\lambda}(\theta C)}{\partial t^{\gamma,\lambda}} = -V\frac{\partial(\theta C)}{\partial x} + D^*\frac{\partial^{\alpha,\kappa}}{\partial x^{\alpha,\kappa}}(\theta C) + q_I C_I - q_o C - \theta r C \tag{13a}$$

$$C(x, t = 0) = \frac{M_0}{\theta}\,\delta(x - x_0) \tag{13b}$$

$$C(x,t)|_{\xi_1} = g_1(t) \tag{13c}$$

$$\left[D^*\frac{\partial^{\alpha-1,\kappa}}{\partial x^{\alpha-1,\kappa}}(\theta C)\right]\Bigg|_{\xi_2} = g_2(t) \tag{13d}$$

$$\left[V\theta C - D^*\frac{\partial^{\alpha-1,\kappa}}{\partial x^{\alpha-1,\kappa}}(\theta C)\right]\Bigg|_{\xi_3} = g_3(t) \tag{13e}$$

where $D^* = \sigma^* V$. Therefore, for 1-$d$ transport with a constant velocity, the scaling factor $\sigma^*$ in the S-FDE is analogous to dispersivity, a parameter commonly used to scale mechanical dispersion (and typically fitted by observed plume data), and the subordination index $\alpha$ is equal to the index of the (tempered) space fractional derivative.

The 1-$d$ adjoint FDE (13) is a simplified version of the 3-$d$ adjoint S-FDE (11):

$$b\frac{\partial(\theta A)}{\partial s} + \beta\frac{\partial^{\gamma,\lambda}(\theta A)}{\partial s^{\gamma,\lambda}} = V\frac{\partial(\theta A)}{\partial x} + D^*\theta\frac{\partial^{\alpha,\kappa}}{\partial(-x)^{\alpha,\kappa}}A - (q_I + \theta r)A + \frac{\partial h}{\partial C} \tag{14a}$$

$$A(x,s)|_{s=0} = 0 \tag{14b}$$

$$A(x,s)|_{\xi_1} = 0 \tag{14c}$$

$$\left[A\theta V - D^*\theta\frac{\partial^{\alpha-1,\kappa}}{\partial(-x)^{\alpha-1,\kappa}}A\right]\Bigg|_{\xi_2} = 0 \tag{14d}$$





$$\left[D^*\theta \frac{\partial^{\alpha-1,\kappa}}{\partial(-x)^{\alpha-1,\kappa}}A\right]\Big|_{\xi_3} = 0 \qquad (14e)$$

The backward FDE (14) is consistent with that derived by Zhang et al. (2022), validating the 1-*d* simplification of the backward

model (11).

When the factor $b = 1$, the capacity coefficient $\beta = 0$ (meaning no immobile phase or solute retention), and the space index $\alpha = 2$ (representing normal diffusion), the forward S-FDE model (13) reduces to the classical $2^{nd}$-order ADE:

$$\frac{\partial(\theta C)}{\partial t} = -V\frac{\partial(\theta C)}{\partial x} + D^*\frac{\partial^2}{\partial x^2}(\theta C) + q_I C_I - q_o C - \theta r C$$

$$C(x, t = 0) = \frac{M_0}{\theta}\,\delta(x - x_0)$$

$$C(x,t)|_{\xi_1} = g_1(t)$$

$$\left[D^*\frac{\partial}{\partial x}(\theta C)\right]\Big|_{\xi_2} = g_2(t)$$

$$\left[V\theta C - D^*\frac{\partial}{\partial x}(\theta C)\right]\Big|_{\xi_3} = g_3(t)$$

and the corresponding backward model (14) is simplified to:

$$\frac{\partial(\theta A)}{\partial s} = V\frac{\partial(\theta A)}{\partial x} + D^*\theta\frac{\partial^2 A}{\partial x^2} - (q_I + \theta r)A + \frac{\partial h}{\partial C} \qquad (15a)$$

$$A(x,s)|_{s=0} = 0 \qquad (15b)$$

$$A(x,s)|_{\xi_1} = 0 \qquad (15c)$$

$$\left[A\theta V + D^*\theta\frac{\partial A}{\partial x}\right]\Big|_{\xi_2} = 0 \qquad (15d)$$

$$\left[D^*\theta\frac{\partial A}{\partial x}\right]\Big|_{\xi_3} = 0 \qquad (15e)$$

which is the same as the 1-*d* backward ADE derived by Neupauer and Wilson (1999).

The applicability of both the 3-*d* backward model (11) and its 1-*d* simplification (14) is examined using real-world aquifers and streams in **section 3**. The 3-*d* backward model (11) is needed since most transport processes in natural aquifers are multi-dimensional. The 1-*d* backward model (14) can also be useful since (i) many times we need to first focus on longitudinal transport, and (ii) most successful applications of the FDEs in hydrology are limited to 1-*d*; see the extensive review by Zhang et al. (2017). The classical 1-*d* backward ADE model (15) will also be applied to reveal the impact of non-

Fickian transport on BTTP by comparing with the adjoint S-FDE solutions.

### 2.2 Lagrangian solver

The adjoint S-FDE (11) with complex boundary conditions cannot be solved analytically to obtain the BTTP, and hence a grid-free, fully Lagrangian numerical solver is proposed here. The Lagrangian solver for the forward-in-time S-FDE (1) under various boundary conditions was developed and tested by Zhang et al. (2019a). We briefly introduce it here. This

forward-in-time Lagrangian solver contains three main steps. *Step 1* decomposes mobile and immobile phases using the





temporal Langevin equation, which is a stochastic model that separates particle waiting time and operational time whose probability density function (PDF) follows the tempered stable density with index $\gamma$ (Meerschaert et al., 2008). *Step 2* applies subordination to regional flow by calculating the streamline-oriented random mechanical displacement for each particle (whose PDF follows the tempered $\alpha$-stable density) rescaled by the local velocity. *Step 3* then adjusts particle trajectories around

boundaries according to the particle-tracking schemes developed by Zhang et al. (2015).

We convert the above-mentioned forward-in-time Lagrangian solver to its backward counterpart to approximate the adjoint S-FDE (11), by incorporating three main modifications. *First*, each vector component of the velocity is reversed to calculate the backward advective displacement of particles during the operational time. *Second*, the skewness of the (tempered) $\alpha$-stable Lévy jumps is changed from the positive maximum (to capture downward mechanical displacement) to the negative

maximum (to backtrack pollutants located upstream initially). *Third*, the source/sink terms and boundary conditions are modified according to those defined in the adjoint model (11) and **Table 1**. For example, the sink term in the forward model, which is $-q_o C$ in Eq. (1a), is replaced by the load term $\frac{\partial h}{\partial c}$ in the adjoint model (11a) which describes the initial probability source in the backward Lagrangian solver. **Table 1** shows the main changes and hydrogeologic interpretations of these boundary conditions (including their value and type) converted from the forward S-FDE to its backward counterpart at the

upstream (inlet) and downstream (outlet) boundaries. For simplicity, **Table 1** uses the 1-$d$ simplification and assumes that the forward flow direction is from left to right. The Dirichlet, Neumann, Robin, and infinite boundaries in the forward model transform to the absorbing, fully reflective, partially reflective, and free boundaries in the backward model, respectively, to correctly backtrack particle trajectories around boundaries and recover pollutant release history. For example, the non-zero, Dirichlet boundary condition in the forward model (expressed by Eq. (1c)) converts to an absorbing boundary in the backward

model (expressed by Eq. (11c)), which is expected since the source term in the forward model becomes the sink term in the backward model. In addition, a non-zero, Neumann boundary condition in the forward model (1-$d$) (representing an immobile diffusive source located at the inlet boundary) transforms to a fully reflective boundary condition in the backward model (11d) (meaning that no external sources outside the upstream boundary), which is necessary to ensure that no particles can exist this upstream boundary (**Table 1**).

This backward-in-time Lagrangian solver is significantly more computationally efficient than the standard Eulerian solver, since (i) particles in the immobile phase remain motionless and therefore do not require any calculation, and (ii) the streamlines can be calculated semi-analytically (LaBolle, 2006) for the streamline-projected mechanical dispersion during subordination to regional flow.

### 2.3 Numerical experiments and validation

Here we check this Lagrangian solver using either simple cases (i.e., 1-$d$) or qualitative evaluation due to the lack of other numerical solvers for the 3-$d$ adjoint S-FDE (11a). The number density of particles exiting the source location (re-scaled





by the velocity) defines the flux-concentration based BTTP, which estimates the PDF of each release time ($s$) for the pollutants identified at the monitoring well at present.

Results of the first numerical experiments are plotted in **Figure 1**. For validation, an implicit Eulerian finite difference

solver of the 1-$d$ adjoint FDE (14a) was also developed by adopting the Grünwald approximation scheme proposed by Meerschaert and Tadjeran (2004) which can efficiently approximate fractional derivatives. The Lagrangian solutions of BTTPs, although containing apparent noise at low BTTPs due to the finite number of particles used in the model, generally match the Eulerian solutions (**Figure 1**). Here the backward travel distance, which represents the known source location, is assumed to be 10 (dimensionless), and the model domain dimension is 100 times larger than the backward travel distance. The

boundaries therefore can be assumed to be infinite, and hence, the free boundary condition listed in **Table 1** is applied for the Lagrangian solver. Numerical analysis also reveals that, when all the other parameters remain unchanged, a larger time truncation parameter $\lambda$ delays the peak time of the BTTP (because a larger $\lambda$ leads to a longer peak waiting time in the truncated stable density) and shrinks the late-time tail of the BTTP (because a larger $\lambda$ significantly narrows the particle's waiting time PDF by truncating extremely long waiting times) (**Figures 1a** and **1b**). In addition, when $\lambda$ is very small (i.e., $\lambda \leq 10^{-6} \ T^{-1}$,

representing an untruncated, standard stable density for the random waiting time), the late-time BTTP tail declines at a rate of $s^{-1-\gamma}$ (**Figure 1d**). When the space truncation parameter $\kappa$ is small and negligible, the early-time BTTP tail increases at a rate of $s^1$ (this "+1" slope in a log-log plot remains stable for all the subordination index $\alpha$ varying from 1 to 2) (**Figures 1b** and **1d**). When all the other parameters remain unchanged, a smaller subordination index $\alpha$ and a larger time index $\gamma$ accelerates the BTTP peak, because a smaller $\alpha$ engenders a faster-moving plume peak and a larger $\gamma$ describes weaker retention.

Therefore, the BTTP early-time tailing behaviour (representing super-diffusion) is dominated by the subordination index $\alpha$ and its truncation parameter $\kappa$, the BTTP late-time tailing behaviour (representing sub-diffusion) is mainly controlled by the time index $\gamma$ and its truncation parameter $\lambda$, and the BTTP peak is affected by all these four parameters (due to the competition between super- and sub-diffusive transport). These BTTP features can be critical signals for real-world applications. For example, the BTTP peak time describes the most likely release time of an instantaneous point source, and the BTTP tails

control the backward travel time distribution which also defines the groundwater age distribution (see the application in **section 3.2**) and transient indexes for assessing aquifer vulnerability (Zhang et al., 2018).

The second numerical experiments apply the Lagrangian solver to backtrack particles in non-uniform flow fields (**Figure 2**). Two 2-$d$ Brownian random hydraulic conductivity ($K$) fields were first generated using the method developed by Zhang et al. (2019a) (**Figures 2a** and **2c**). Steady-state groundwater flow was then calculated by the United States Geological Survey

(USGS) software MODFLOW (Harbaugh, 2005) (shown by the black lines in **Figures 2b** and **2d**). Backward particle tracking plumes were finally obtained by the Lagrangian solver proposed above (shown by the contour map in **Figures 2b** and **2d**). For $K$ field #1 with a relatively "homogeneous" distribution of $K$, particles starting from different wells move backward at a similar rate and are eventually removed from the system when reaching the upstream boundary (which is the left boundary located at $x = 0$ and is assumed to be an absorbing boundary in the backward model) (**Figure 2b**). All plumes follow the general path





of local streamlines, as expected for the streamline projection method proposed above. The transverse expansion of the plume

is due to molecular diffusion added to particle dynamics (to capture hydrodynamic dispersion). For *K* field #2 representing a

more heterogeneous *K* field with layering deposits, particles starting from the high-*K* zone move quickly and then exit the

model domain (**Figure 2d**). These backward dynamics follow our logical expectations, but cannot be validated since, to the

best of our knowledge, there are no other solvers available for the vector model (11).

**3 Field applications**

The adjoint S-FDE model is applied in this section to recover the release history of pollutants in aquifers and rivers and

calculate groundwater ages dated by environmental tracers. As shown below, these surface and subsurface flow systems exhibit

different degrees of medium heterogeneity, various flow velocity resolutions and boundary conditions, and different

spatiotemporal scales, which provide an ideal set of natural variability to test the real-world applicability of the physical model

and numerical solver developed in this study.

**3.1 BTTP application case 1: Recover release history of pollutants at the MADE site**

Natural-gradient tracer tests were conducted at the Macrodispersion Experiment (MADE) site in Columbus, Mississippi,

U.S. (Adams and Gelhar, 1992; Boggs et al., 1992), identifying mixed sub- and super-diffusive pollutant transport in a ~11 m

thick and ~300 m long alluvial aquifer (Bianchi et al., 2016; Yin et al., 2020). Non-Fickian transport at the MADE site

motivated the development of various numerical/stochastic transport models in the last three decades (see the review by Zheng

et al. (2011)), but the BTTP dominated by mixed sub/super-diffusion remained unknown. Here we calculate its BTTP using

the adjoint S-FDE (14a) (which is an upscaled model) with a uniform velocity. The 1-*d* backward model is selected since the

MADE site transport can be simplified by a 1-*d* process projected to the longitudinal direction, as demonstrated by many

previous models (Zheng et al., 2011).

The seven parameters in the backward model (14a) can be conveniently estimated using mainly literature data. The strong

sub- and super-diffusion observed at the MADE site implies that the two truncation parameters ($\lambda$ and $\kappa$) can be simply

neglected, leaving 5 unknown parameters in model (14a). The subordination index $\alpha$ is analogous to the spatial index (=1.1)

estimated by Benson et al. (2001) using the distribution of measured permeability. The time index $\gamma$ (=0.39) and capacity

coefficient $\beta$ (=0.082 day$^{\gamma-1}$) were estimated by Zhang et al. (2010) using the decline rate of the observed mobile tracer mass.

The velocity $V$ (=0.24 m/day) can be approximated by the mean velocity measured in the field, and the scaling factor $\sigma^*$ is

assumed to be 1 m since dispersion at the MADE site was found to be the same order as $V$ (Benson et al., 2001).

The predicted BTTPs are plotted in **Figure 3**. Here we choose the monitoring well located at the bromide plume peak

(obtained in the MADE-1 bromide tracer test) as the detection location denoted as $x_w$ (which is defined as the location of the

monitoring well detecting the maximum concentration), since this location represents the mass center of the tracer plume. The





contaminant source was known at the origin ($x_0 = 0$). The plume peak at the first (Day 49) and second (Day 126) sampling

cycles is located at $x_w = 3.0$ m and 7.0 m, respectively, providing two possible detection locations. These two detection

locations lead to the two predicted BTTPs depicted in **Figure 3**, after applying the adjoint S-FDE (14a) with the seven

parameters estimated above.

The model results show that, on the one hand, the peak of the flux-concentration based BTTP captures well the true

release time (**Figures 3a** and **3b**). On the other hand, the peak of the BTTP based on the concentration profile for "immobile"

particles (which were located at the source location and remained motionless during each unit time interval in calculating

BTTP), has a higher value and corresponds to a much later time (twice of the flux-concentration based BTTP peak), which

significantly overestimates the true release time. This discrepancy is explained by the immobile phase source moving slower

than the mobile phase source due to strong solute retention, resulting in an older release time. For an aqueous phase observation,

the flux-concentration based BTTP describes the PDF of release time for aqueous (or mobile) phase sources, while the

immobile particles' concentration based BTTP describes the PDF of release time for absorbed (or immobile) phase sources.

In the MADE-1 tracer test, the bromide tracer was injected into the upstream well as an initially mobile source, and hence the

flux-concentration based BTTP is needed, meaning that the adjoint S-FDE (14a) successfully recovers the tracer release

history. In addition, as shown in **Figure 3c**, the slope of the late-time BTTP for the immobile phase sources in a log-log plot

(which is $-\gamma$) is -1 smaller (i.e., heavier) than that for the mobile phase sources (which is $-\gamma - 1$), describing the persistent

release of immobile pollutant mass at the source location and implying a high degree of uncertainty in BTTP for the immobile

phase source.

The adjoint ADE is also applied here for comparison. When the same velocity $V$ (=0.24 m/day) and dispersion coefficient

$D^*$ ($= \sigma^* V = 0.24$ m²/day) are used, the adjoint ADE significantly underestimates the true release time (not shown here). This

is expected because the adjoint ADE cannot capture solute retention. We then calibrate $V$ (=0.068 m/day) and $D^*$ (=0.68

m²/day) by fitting the mean and variance of the observed bromide plumes, but the resultant BTTP peak still underestimates

the true release time by more than one order of magnitude (shown by the solid black line in **Figure 3**). Finally, we directly fit

$V$ (=0.026 m/day, which is one order of magnitude smaller than the mean groundwater velocity) and $D^*$ (=0.031 m²/day) using

the true release time for the detection well located at $x_w = 3.0$ m (shown by the dashed black line in **Figure 3a**), but the best-

fit adjoint ADE then overestimates the true release time by > 50% for the detection well at $x_w = 7.0$ m (shown by the dashed

black line in **Figure 3b**). Therefore, the adjoint ADE cannot reliably recover the release history of pollutants undergoing strong

non-Fickian transport in the MADE aquifer. The same conclusion was drawn by previous studies for fitting tracer transport at

the MADE site using the ADE based models (Zheng et al., 2011).

**3.2 BTTP application case 2: Groundwater age dating in Kings River alluvial aquifer, California**

The vector backward S-FDE (11a) is then used to calculate the distribution of groundwater ages at the Kings River

alluvial aquifer (KRAA) located in Fresno County, California, U.S. (**Figure 4**). The flux-concentration based BTTP also





represents the groundwater age distribution and provides core information for groundwater sustainability assessment (Fogg et al., 1999; Weissmann et al., 2002; Fogg and LaBolle, 2006).

The KRAA system consists of five paleosol-bounded stratigraphic sequences recognized by Weissmann and Fogg (1999). One realization of the 3-*d* hydrofacies model built upon the Markov Chain model developed by Weissmann et al. (2004) is shown in **Figure 4**, where the hydrofacies model contains both the large-scale stratigraphic sequences and the intermediate-scale hydrofacies within each sequence. This 3-*d* Markov Chain model was built using hydrofacies distribution obtained from 11 cores, 132 drillers' logs, and soil survey data. The hydrofacies model contains all the cores and drillers' logs as hard conditional data, to maximally incorporate observed information into the numerical model. This regional-scale model

contains ~1 million cells, with the cell's dimension of 200 m, 200 m, and 0.5 m in the depositional strike, depositional dip, and vertical directions, respectively, with a total model domain size of 12,600 m × 15,000 m ×100.5 m along these three directions. The steady-state groundwater flow was then calculated by MODFLOW using parameters/boundary conditions described by Weissmann et al. (2004) and Zhang et al. (2018b). For example, the measured $K$ was assigned to each facies (gravel, sand, muddy sand, mud, and paleosol). The top of the model accounted for a recharge boundary, and the lateral and basal boundaries

of the model were general head boundaries to allow inflow/outflow. The modeled hydraulic heads were close to the measured data (Zhang et al., 2018b). The resultant fine-resolution velocity field was used to calculate BTTP using the adjoint S-FDE (11a).

We first conduct a parameter sensitivity test using the adjoint S-FDE (11). In these backward particle tracking models, both the water table (representing an internal boundary) and the lateral, upstream boundary of the model are set as absorbing

boundaries (because they represent the source locations), and the other model boundaries are simply treated as fully reflective boundaries. An effective porosity of 0.33, which was the best-fit value in Weissmann et al. (2004) and Zhang et al. (2018b), is applied for these simulations. Three cases with decreasing super-diffusion and increasing sub-diffusion are considered here. Case 1 captures strong super-diffusion (with the time index $\gamma = 0.80$, the capacity coefficient $\beta = 0.1$ yr$^{\gamma-1}$, the subordination index $\alpha = 1.40$, and the scaling factor $\sigma^* = 0.4$ m), Case 2 represents the intermediate scenario (with $\gamma = 0.72$, $\beta = 0.2$ yr$^{\gamma-}$

$^1$, $\alpha = 1.45$, and $\sigma^* = 0.3$ m), and Case 3 describes strong sub-diffusion (with $\gamma = 0.65$, $\beta = 0.3$ yr$^{\gamma-1}$, $\alpha = 1.50$, and $\sigma^* = 0.2$ m). The subordination truncation parameter remains the same for all three cases ($\kappa = 1.0 \times 10^{-5}$ m$^{-1}$). The resultant backward particle tracking snapshot at the backward time $s$=50 yrs is plotted in **Figures 5a~5c** for these three cases. Driven by subordination to regional flow, particles move along streamlines and expand especially in high-permeability deposits (due also to molecular diffusion simultaneously along all three axis directions). Case 1 captures fast backward (i.e., toward

upstream) movement of particles due to strong super-diffusion, such that most particles arrive at the water table within 50 yrs and are then removed from the system, leaving a few particles behind (**Figure 5a**). Contrarily, Case 3 captures the most delayed backward movement due to strong sub-diffusion, and most particles remain in the aquifer with a limited spatial expansion, as shown in **Figure 5c**. This parameter sensitivity test, therefore, shows that the adjoint S-FDE (11) can reasonably interpret non-Fickian dynamics in multi-dimensional aquifers. In addition, the corresponding BTTP for each case, which represents the age


distribution for groundwater sampled at the well screen shown in **Figure 5a** (the green rectangle), is plotted in **Figure 5d**.
With a larger subordination index $\alpha$ and a smaller time index $\gamma$ in the adjoint S-FDE (i.e., from Case 1 to Case 3), the BTTP
shifts apparently toward older ages with a decreasing peak and an expanding distribution, characterizing the impact of
decreasing super-diffusion and increasing sub-diffusion on groundwater age distributions. This test shows that main properties
of the BTTP, including the mean, peak, and variance of groundwater ages, are sensitive to the two indexes $\alpha$ and $\gamma$. Further

comparisons show that the classical adjoint ADE misses the early arrivals of the BTTP, because it cannot capture super-
diffusion (figures not shown).

Finally, the adjoint S-FDE solutions were compared to chlorofluorocarbon-11 (CFC-11) ages measured by Burow et al.
(1999) from USGS for KRAA in 1994. The S-FDE model parameters cannot be predicted using the hydrofacies property-
based method proposed by Zhang et al. (2014) for stationary hydrofacies models, due to the nonstationary distribution of

hydrofacies at KRAA. An alternative (of parameter fitting) is the age distribution for groundwater, especially for shallow
groundwater, which can be calibrated using the groundwater age dated by environmental tracers such as CFCs. **Figures 6a~6d**
show the calculated BTTP for the USGS wells sampled by Burow et al. (1999) (listed in **Figure 4**). Both the adjoint S-FDE
(11a) and the adjoint ADE (15a) were first calibrated to fit the measured CFC-11 age of Well B41 (the modeling of CFC ages
followed the methodology proposed by Weissmann et al. (2002)). Preliminary tests showed that the simulated CCF-11 age is

insensitive to the two truncation parameters, since the subordination truncation parameter $\kappa$ (or the temporal truncation
parameter $\lambda$) mainly affects the very early time, i.e., $< 1$ day (or very late time, i.e., $> 50$ yrs) in the BTTP. The velocity field
was resolved directly from the MODLFOW solutions of hydraulic head, and therefore velocity was not a fitted parameter.
Hence, the adjoint S-FDE (11a) now has 4 unknown parameters: the subordination index $\alpha$ and the scaling factor $\sigma^*$ which
control the climbing limb of the BTTP, and the time index $\gamma$ and the capacity coefficient $\beta$ which control the declining limb

of the BTTP. The competition between these two groups of parameters (mainly the two indexes) affects the BTTP peak, as
discussed in **Section 2.3**. Here the primary goal is to select the best-fit set of parameters while remaining within the known
range for these two indexes which define super- and sub-diffusion. To capture strong super-diffusion within a very coarse
velocity field such as a uniform velocity, the subordination index $\alpha$ ($1 < \alpha \leq 2$) should be close to the lower end (for example,
the MADE-1 site has a best-fit $\alpha = 1.1$ when a uniform, upscaled velocity is used); similarly, for modeling strong sub-

diffusion with a uniform velocity, the time index $\gamma$ ($0 < \gamma \leq 1$) needs to be close to the lower end (for example, the MADE-
1 site has a best-fit $\gamma = 0.39$). With an increase in the resolution of velocity, values of $\alpha$ (or $\gamma$) increase and may approach the
upper limit of 2 (or 1) if velocity is resolved at the pore-scale. The fine-resolution velocity field available for KRAA allow for
the selection of $\alpha$ and $\gamma$ close to their upper ends in trial-and-error calibrations, leading to the following best-fit results: the
subordination index $\alpha = 1.90$, the scaling factor $\sigma^* = 0.2$ m$^{-1}$, the time index $\gamma = 0.80$, and the capacity coefficient $\beta = 0.2$

day$^{\gamma-1}$. For the adjoint ADE, the only fitting parameter is dispersivity, and the best-fit isotropic dispersivity (longitudinal and
transverse dispersivities $\alpha_L$ and $\alpha_T$) is 0.04 m. This same value of isotropic dispersity was also applied by previous studies for
modeling KRAA transport processes using ADE based models by Weissmann et al. (2002, 2004) and Zhang (et al. , 2018b),





who found that (i) simulation results were not sensitive to the value of $\alpha_L$ (because plume spreading is mainly controlled by the hydrofacies-scale heterogeneity captured by the geostatistical model), and (ii) the Lagrangian solver ran faster for isotopic

dispersivity.

The best-fit parameters were then applied to predict the CFC-11 age for the other wells. The CFC-11 age calculated by the adjoint S-FDE matches the observed age better than the adjoint ADE for all wells considered here. The BTTP simulated by the adjoint ADE exhibits multiple or secondary peaks whose locations can differ significantly from the measured CFC-11 age. The adjoint S-FDE, however, usually shows a single peak in the BTTP which is closer to the true CFC-11 age, which may

provide a convenient interpretation of the environmental tracer dating: this tracer-dated apparent age is usually located around (i.e., in the range of the 25[th] to 75[th] percentiles of) the BTTP peak. In addition, **Figure 6e** shows the joint BTTP for all wells, which represents the groundwater recharge time for all four wells simultaneously. The joint BTTP depicted in a log-log plot (**Figure 6j**) is narrower than each marginal BTTP, because the uncertainty (in recovering the pollutant release history) decreases when concentration data from multiple observation wells are available. Notably, this is the first validated large-scale

transport model that combines non-local super/sub-diffusion and local velocities. This application proves the applicability of the adjoint S-FDE (11a) and its Lagrangian solver in capturing BTTP in a 3-$d$, regional-scale, nonstationary alluvial aquifer with a fine-resolution velocity field.

**3.3 BTTP application case 3: Recover the release time for tracers in Red Cedar river, Michigan**

Phanikumar et al. (2007) released fluorescein dye in the Red Cedar River (RCR), a fourth-order stream in Michigan, US,

and then measured the breakthrough curves (BTCs) at three locations with travel distances of 1.4 km, 3.1 km, and 5.08 km, respectively, to explore the impact of river system retention on dissolved chemicals. The resultant BTCs were fitted by Chakraborty et al. (2009) using the standard, 1-$d$ space FDE with a constant velocity. The 1-$d$ model was applicable because of the relatively straight reach. Since sub-diffusion was found in this stream (Phanikumar et al., 2007) (likely due to open channel retention and/or hyporheic exchange) and the space FDE cannot model sub-diffusion, we apply the backward FDE

(14a), which is a more general model (containing both space and time fractional derivatives) than the adjoint of the standard space FDE, to predict the tracer release time.

We first estimated the seven parameters in the 1-$d$ adjoint S-FDE (14a) from the tracer data. The tracer BTCs measured by Phanikumar et al. (2007) all exhibited an exponential mass increase in the BTC's climbing limb and fast mass decrease in the declining limb, implying Fickian diffusion in the operational time (meaning that the subordination index $\alpha$ is close to 2

and the spatial truncation parameter $\kappa$ is negligible) and weak solute retention (so that the time index $\gamma$ should be large, and we selected 0.9 as the initial try). The capacity coefficient $\beta$ should be small, considering ~90% of the mass recovery rate in the field (Phanikumar et al., 2007), and hence we approximated $\beta = 0.08$ minute$^{1-\gamma}$ (representing 90% of mobile mass recovery). The temporal truncation parameter $\lambda$ (=0.034 minute$^{-1}$) was approximated by the reverse of the time interval from the BTC peak to the inflection point of the BTC slope, as shown by Zhang et al. (2022). The mean velocity $V$ (=0.0317





km/minute) was estimated by the speed of the BTC peak moving from the 1$^{st}$ sampling location ($L$=1.4 km) to the 2$^{nd}$ one ($L$=3.1 km). The last parameter, dispersion coefficient $D^*$ ($= \sigma^* V$), was estimated to be 0.00317 km$^2$/minute by assuming that dispersion is one order of magnitude smaller than advection (since solute transport in rivers is usually dominated by advection). These rough estimations contain high uncertainty, but they significantly simplify the field application of a complex model containing 7 unknown parameters.

The peak of the predicted flux-concentration based BTTPs using the 1-$d$ adjoint S-FDE (14a) can capture the true release time for the stream gauges located at $L$=3.1 km (gauge #2) and 5.08 km (gauge #3) (shown by the red solid line in **Figures 7b** and **7c**), although it slightly underestimates the true release time for gauge #1 located at $L$=1.4 km (**Figure 7a**) (this deviation is because the velocity was estimated using the transport data for tracers passing gauge #1). For comparison purposes, the adjoint ADE model is also used here: when the same $V$ (=0.0317 km/minute) and $D^*$ (=0.00317 km$^2$/minute) as those in the S-

FDE are used, the adjoint ADE model underestimates all the true release times (see the black solid line in **Figure 7**). We then fit $V$ and $D^*$ for the first gauge by matching the true release time for tracers captured at gauge #1, but the adjoint ADE model then underestimates the true release time for tracers captured at gauges #2 and #3. Therefore, the adjoint S-FDE (14a) is more appropriate than the classical adjoint ADE for recovering pollutant release history in this river with a constant velocity.

It is also noteworthy that the BTTP for the immobile phase sources has the similar peak time and tailing behaviour as

those in the BTTP for the mobile phase sources (**Figure 7**). This similarity is due to the weak solute retention captured by the large time index $\gamma$ (meaning a relatively narrow distribution of the waiting time PDF), the small capacity coefficient $\beta$ (meaning a smaller portion of immobile pollutants at equilibrium), and the relatively large time truncation parameter $\lambda$ (meaning that pollutant transport approaches Fickian scaling once time exceeds $\frac{1}{\lambda} \approx 32$ minutes). This result differs from that found for the MADE aquifer discussed in **section 3.2**, implying stronger sub-diffusion in regional-scale alluvial

aquifer/aquitard systems than rivers.

## 4. Discussion: Extension of field applications and model capabilities

The adjoint subordination approach developed and applied above can also be used to identify the pollutant source location, which plays a crucial role in pollution source control and water resource management. The backward-in-time vector model (11a) may also be extended to a more general form for more complex transport. These possible extensions are discussed

in the following two subsections.

### 4.1 Identify pollutant source location using backward location probability density function (BLP)

Pollutant source location identification has remained a hot topic in hydrology for more than two decades, as extensively reviewed by Atmadja and Bagtzoglou (2001), Chadalavada et al. (2011), and Moghaddam et al. (2021). Process-based and statistical models had also been developed in the last two years to successfully identify pollutant source in groundwater and





rivers, including genetic algorithms combined with groundwater models (Han et al., 2020; Habiyakare et al., 2022) or optimization models (Ayaz et al., 2022), the modified export coefficient model combined with SWAT (Guo et al., 2022), physical/stochastic inverse models (Moghaddam et al., 2021), isotope mixing models (Wiegner et al., 2021; Ren et al., 2021), deep learning models (Kontos et al., 2021; Pan et al., 2021), the model-based backward probability method (Khoshgou and Neyshabouri, 2022), and the Null space Monte Carlo stochastic model (Pollicino et al., 2021), among many other models.

The adjoint S-FDE (11) provides a new process-based modeling approach in pollutant source location identification. It calculates backward location probability density function (BLP) (which is analogous to the normalized resident concentration at a previous time), where the peak BLP defines the most likely point source location. As shown in **section 3** in recovering pollutant release history, the adjoint S-FDE (11) may improve the classical process-based pollutant source identification models by i) identifying the source location for pollutants undergoing non-Fickian diffusion (including super-diffusion, sub-diffusion, their mixture, and any intermediate transition from non-Fickian to Fickian diffusion), ii) distinguishing the initial source phase, and iii) incorporating flow fields with various resolutions. We check this hypothesis using real-world data below.

**4.1.1 BLP application case 1: SHOAL test site**

The adjoint S-FADE (11a) was first applied to identify the point tracer source at the SHOAL test site, Churchill County, central Nevada, US. A radial tracer test was conducted by Reimus et al. (2003) in a saturated, fractured granite located at the SHOAL site. The detailed fracture configuration was not available for the granite aquifer, although researchers classified the discrete fracture networks (DFNs) into three groups (small, medium, and large, according to the fracture aperture) using a stochastic approach (Pohll et al., 1999). The slow, ambient groundwater velocity was estimated to be 0.3 to 3 m/yr (Pohll et al., 1999), which was negligible compared to the radial flow generated by the pumping test. A total of 20.81 kg of bromide with an average concentration of 3.6 g/L was injected into an injection well located 30 m from the extraction well. The measured tracer BTC exhibited both early time and late time power law tails, although the late time BTC was too short to reveal the exhaustive mass decline (see symbols in **Figure 8**).

We applied MODFLOW to calculate the steady-state flow, by simplifying the complex velocity field as a radial flow with an average pumping rate of $Q = 12.4$ m³/day (the same value used for the SHOAL field test). The simplified "homogeneous" aquifer (selected here for the purpose of upscaling) has an average $K$ of 5.78×10⁻⁶ m/s (in the range of the bulk hydraulic conductivity, which was $1.48×10^{-6} \sim 4.7× 10^{-5}$ cm/s, measured by Pohll et al. (1999)). The vector S-FDE (1a) with a convergent flow field was then applied to fit the observed bromide BTC. **Figure 8** compares the measured and fitted bromide BTCs. The best-fit parameters in the S-FDE model (1a) are as follows: the time index $\gamma = 0.44$ (without truncation), the capacity coefficient $\beta = 0.48$ d$^{\gamma-1}$, the subordination index $\alpha = 1.95$, the scalar factor $\sigma^* = 1.0$, the truncation parameter $\kappa = 1.3 \times 10^{-3}$ m⁻¹, and the molecular diffusion coefficient $D^* = 1.0 \times 10^{-5}$ m²/d. The resultant 2-$d$ forward-in-time plume snapshots (along the horizontal plane) using model (1a) are plotted in **Figure 9** at both early time ($t = 2$ d) and late time ($t = 200$ d) for all phases (the mobile, immobile, and total phases). The simulated fractional mass recovery at the last sampling



cycle ($t = 322$ d) for the tracer bromide was 20.2%, which is close to the recovery ratio (18.0%) estimated by Reimus et al. (2003).

The resultant backward streamlines using the adjoint S-FDE (11a) are perpendicular to the groundwater head contour (**Figure 10a**), validating the concept of subordination to regional flow and our Lagrangian solver: particles should move backward along streamlines, to describe the backward mechanical dispersion. The simulated BLP is plotted in **Figures 10b~10d**, where the peak BLP for the mobile phase source captures the true point source location (note that the initial point source was in the mobile phase), while the peak BLP for the immobile phase source stays behind and is closer to the pumping well (due to strong retention). The adjoint S-FDE (11a) and its Lagrangian solver developed in **section 2**, therefore, can

calculate the BLP for a divergent flow field in a 2-*d* fractured aquifer.

### 4.1.2 BLP application case 2: KRAA

We then apply the adjoint S-FDE (11a) to calculate BLP for non-point pollutant sources for the KRAA aquifer. **Figure 11a** shows the resultant BLP for well B51. Here the BLP captures properties (i.e., locations and weights) of non-point source pollutants from the water table that can reach Well 51 during the last 200 yrs. It can also be adopted as the well head protection

zone (under the ambient flow condition, i.e., without pumping). To explore the sensitivity of BLP to the well depth, a deeper well named "5b" (which is 14.0 m deeper, right below Well 51) was also modeled, with the resultant BLP plotted in **Figure 11b**. The BLP for Well 5b exhibits a relatively closer source center than that for Well 51, implying the existence of preferential flow paths at the deeper aquifer which can be captured by the adjoint S-FDE (11a). **Figure 11c** shows the joint BLP for wells 51 and 5b, identifying the locations for non-point source pollutants that can contaminate both wells. For comparison purposes,

the BLP was also calculated by the adjoint ADE, which covers a larger area especially near the monitoring well (**Figure 11d**), which is most likely due to the strong transverse (vertical) dispersivity ($\alpha_T = 0.04$ m) (see **section 3.2**). With the increase of well depth, the center of the related pollutant sources moved further upstream (**Figure 11e**). Overall, the majority of BLP calculated by the adjoint S-FDE (11a) is located inside of the BLP calculated by the adjoint ADE (**Figure 11g**), implying that the adjoint S-FDE (11a) tends to reduce the uncertainty in pollutant source identification by emphasizing the impact of

dominant flow paths (including the preferential flow paths) on regional-scale pollutant transport. This also explains why the BLP calculated by the adjoint S-FDE extended slightly further upstream than that of the adjoint ADE (because the adjoint S-FDE captures super-diffusive, large-scale jumps).

### 4.2 Extension to multi-scaling subordinated model

The backward-in-time vector model (11a) has two main limitations. First, it requires up to seven parameters whose

predictability remains a challenge. This study provided preliminary tests for model parameter estimation (in **sections 3 and 4**), and further parameter predictability for fractional-derivative models can be found in Zhang et al. (2022). More efforts are still needed in future studies to improve the predictability of FDEs.


Second, the subordination index $\alpha$ and scaling factor $\sigma^*$ in model (11a) are limited to constant values, while pollutant plumes in natural geological media may exhibit non-uniform, super-diffusive spreading rates. As a preliminary test, here we propose the following multi-scaling subordination model as a possible extension of (11a), by adopting the multi-scaling fractional derivative concept proposed by Meerschaert et al. (2001):

$$b\frac{\partial(\theta A)}{\partial s} + \beta\frac{\partial^{\gamma,\lambda}(\theta A)}{\partial s^{\gamma,\lambda}} = \nabla_{\bar{V}}(\theta A) - \theta(\nabla_{\bar{V}})^{\mathbf{H}(\bar{V})^{-1}}_{M(\bar{V})}A - (q_I + \theta r)A + \frac{\partial h}{\partial C} \, , \tag{16}$$

where $M(\bar{V})$ denotes the mixing measure which defines the (rescaled) probability for particles moving along each direction of the vector velocity $\bar{V}$, and $\mathbf{H}(\bar{V})^{-1}$ denotes the inverse of the scaling matrix which defines the subordination index (with tempering) along the water flow direction of $\bar{V}$. When $M(\bar{V})$ is constant (i.e., reduces to the constant $\sigma^*$) and the matrix $\mathbf{H}(\bar{V})^{-1}$ reduces to a constant $\bar{\alpha}$ (with the truncation parameter $\kappa$) along all directions, the multi-scaling adjoint S-FDE (16) reduces to the unique-scaling model (11a).

The general model (16) allows direction-dependent scaling rates for capturing multi-dimensional transport in complex media such as regional-scale fractured media. This function is similar to the multi-scaling adjoint fractional-derivative model derived by Zhang (2022):

$$b\frac{\partial(\theta A)}{\partial s} + \beta\frac{\partial^{\gamma,\lambda}(\theta A)}{\partial s^{\gamma,\lambda}} = \nabla \cdot (\theta\vec{V}A) - \theta D\, \nabla^{\bar{\mathbf{H}}^{-1}}_{\bar{M}(d\theta)}A - (q_I + \theta r)A + \frac{\partial h}{\partial C} \, , \tag{17}$$

where the mixing measure $\bar{M}(d\theta) = M(d\theta + \pi)$ is reversed for each discrete angle $d\theta$ for backward particle jumps, and the corresponding scaling matrix $\bar{\mathbf{H}}$ is also reversed by $\pi$ along each eigenvector direction. The multi-scaling adjoint FDE (17) applies for a space-dependent velocity vector $\vec{V}$, where the spreading angles and weights in the mixing measure $\bar{M}(d\theta)$ can change with velocity. The computational burden of model (17), however, increases with an increasing resolution of the flow field, because the particle displacement during each jump event needs to be separated into multiple sections and then projected into the adjacent streamline deviating with the angle of $d\theta + \pi$ from the starting velocity vector (which can be called the streamline projection method with non-zero projection angles), as demonstrated by Zhang (2022). This can lead to prohibitive computational burden for a regional-scale aquifer with complex flow, such as the KRAA site. The multi-scaling adjoint S-FDE (16) solves this challenge using the streamline-orientation approach, meaning that there is no need to deviate by an angle of $d\theta + \pi$ because mechanical dispersion follows the streamlines.

Here we first check the Lagrangian solution of model (16) for a simple case where the other solution is available. **Figure 12c** shows the Lagrangian solution of the multi-scaling S-FDE, given the mixing measure (with divergent flow) and the scaling matrix (with a constant index) shown in **Figure 12b**. This may define pollutant transport in a discrete fracture network (DFN) with multiple orientations (**Figure 12a**). The Lagrangian solution matches well Nolan's (1998) multivariate stable distribution (**Figure 12d**).

Next, we apply model (16) to track pollutant transport in a 2-$d$ DFN. **Figure 13a** shows the ensemble average of plume snapshots at time $t$=4.6 yrs from Monte Carlos simulations of pollutant transport in 100 DFNs generated by Reeves et al.





(2008), where the DFN has multiple orientations, and the plume therefore moves along various directions. The best-fit solution
using the forward-in-time, multi-scaling S-FDE is shown in **Figure 13c**, which can capture the fingering of the plume due to
super-diffusion along fractures. For comparison purposes, we also apply the multi-scaling FDE proposed by Zhang (2022) to
capture the plume snapshot (**Figure 13b**), which is similar to that of the multi-scaling S-FDE. The best-fit parameters are then
applied to predict plume snapshots at two later times. The multi-scaling S-FDE can capture the plume's center density and rear
edge slightly better than the multi-scaling FDE (see for example, **Figures 13f** vs. **13g** and **Figures 13j** vs. **13k**). The peak of
the corresponding BLP calculated by the multi-scaling adjoint S-FDE (16) (where the reflective boundary condition is used
for each boundary, since no pollutants recharge from the outside) can capture the true point source location (note that the plume
center did not move apparently downstream due to strong matrix diffusion). Details of model parameter estimation for the
DFNs can be found in Zhang (2022). This application shows that the multi-scaling adjoint S-FDE (16) can conveniently
identify the pollutant source location in DFNs with a uniform, upscaling velocity vector.

**5. Conclusion**

To reliably track pollutants in natural water flow systems, this study derived the adjoint of the time-fractional nonlocal
transport model subordinated to regional flow, developed the fully Lagrangian solver, and then applied the new approach to
track pollutants undergoing non-Fickian transport in surface water and groundwater with various velocity resolutions.
Mathematical analysis and real-world hydrologic applications revealed the following four main conclusions.

First, the adjoint subordination approach led to an adjoint S-FDE model for quantifying backward probabilities, which
takes subordination to the reversed regional flow, converts the forward-in-time boundary conditions, and reverses the tempered
$\alpha$-stable density for mechanical dispersion. The resultant backward-in-time boundary conditions can either capture the outside
pollutant sources using the absorbing/free boundary or exclude any out-of-domain pollutant sources using the fully reflective
boundary (all of these boundary conditions were tested in applications). The adjoint $\alpha$-stable density (with tempering) reverses
its skewness to describe backward, super-diffusive large displacements of particles along preferential flow paths, which is
combined with the self-adjoint time fractional derivative term in the model (for describing sub-diffusion) to capture a wide
range of non-Fickian transport dynamics. In addition, the corresponding Lagrangian solver is computationally efficient because
backward super-diffusive mechanical dispersion of particles can be tracked by simply reversing streamlines.

Second, real-world applications showed that the adjoint S-FDE reliably tracked pollutants moving in surface water and
groundwater with various resolutions of velocity. The new model successfully recovered the release history and identified the
location(s) of pollutant source(s) for water systems with a uniform velocity, a non-uniform flow field (i.e.,
divergent/convergent flow), and fine-resolution velocities in a non-stationary, regional-scale alluvial aquifer. In these
simplified or well-characterized flow fields, non-Fickian dynamics especially sub-diffusion (due to for example solute
retention, hyporheic exchange, or matrix diffusion) were ubiquitous and affecting pollutant transport processes, and the adjoint
S-FDE performed better than the classical ADE based backward models in calculating BTTP and BLP.





Third, caution regarding the pollutant source phase is needed when backtracking pollutants in natural geologic media. For example, the mobile phase pollutant source can exhibit a much shorter release time and an apparently further source location than the immobile phase source in alluvial aquifers where sub-diffusion is typically very strong due to the usually abundant aquitard materials. The mobile-immobile pollutant source phase distinction, however, may be neglected for large-scale transport in rivers with weak solute retention. Field tracer tests (including those revisited in this study) usually had a mobile initial phase, but real-world applications may involve immobile pollutant sources (such as DNAPL) where the method proposed in this study may be applied.

Fourth, field applications of the adjoint S-FDE are challenged by the poor predictability of model parameters, and the model itself may be extended for more complex transport dynamics. This study provided simple estimations for model parameters given field measurements, while future efforts are still needed to link quantitatively model parameters to media/pollutant properties. In addition, the multi-scaling adjoint S-FDE may extend the unique-scaling adjoint S-FDE and simplify the multi-scaling adjoint FDE in tracking pollutants in fractured media.

**Data availability**

Data for BTTP Application 1 are available from the published paper Benson et al., Transport in Porous Media, 2001 at https://link.springer.com/article/10.1023/A:1006733002131. Groundwater age data using CFC-11 are available online from the reference Burow et al., U.S. Geol. Surv. Water Resour. Invest., 1999. SHOAL test site data are available from the published paper Reimus et al., Water Resour. Res. (2003) at https://agupubs.onlinelibrary.wiley.com/doi/full/10.1029/2002WR001597. The discrete fracture network data are available from the published paper Reeves et al., Water Resour. Res. (2008) at https://agupubs.onlinelibrary.wiley.com/doi/full/10.1029/2008WR006858. All the numerical data are available from the Zendo repository (Yong Zhang, 2022).

**Author contributions**

YZ led the investigation, conceptualized the research, did the formal analysis, supervised the project, and wrote the initial draft. HGS acquired the funding and the resources. All co-authors reviewed and edited the paper.

**Competing interests**

The contact author has declared that neither they nor their co-authors have any competing interests.

**Acknowledgments**

HGS was partially funded by the National Natural Science Foundation of China (Grant numbers U2267218 and 11972148). YZ was partially funded by the Department of the Treasury under the Resources and Ecosystems Sustainability, Tourist Opportunities, and Revived Economies of the Gulf Coast States Act of 2012 (RESTORE Act). The statements,





findings, conclusions, and recommendations are those of the authors and do not necessarily reflect the views of the Department
of the Treasury or ADCNR. This paper does not necessary reflect the view of the funding agencies.

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





**Table 1**. Changes of boundary conditions from the 1-$d$ forward FDE (13a) to its backward model (14a).

| Boundary | Forward S-FDE (1a) | Backward S-FDE (11a) |
|---|---|---|
| Left (upstream) | **Dirichlet boundary**: $C\|_{x=L} = g_1(t)$, representing a stagnant source reservoir at the inlet. | **Absorbing boundary**: $A\|_{x=L} = 0$, which can be used for groundwater age modeling (the foreword source term becomes the backward sink term). |
| | **Neumann boundary**: $-\frac{\partial^{\alpha-2}}{\partial x^{\alpha-2}}\left[\theta D \frac{\partial(e^{\kappa x}C)}{\partial x}\right]\Big\|_{x=L} = g_1(t)$, representing an immobile diffusive source located at the inlet (less common). | **Fully reflective boundary**: $\left[-V\theta A + \theta D \frac{\partial^{\alpha-1}(e^{-\kappa}A)}{\partial(-x)^{\alpha-1}} e^{\kappa x}\right]\Big\|_{x=L} = 0$, where no particles can exist this upstream boundary; so, there are no external sources outside the upstream boundary. |
| | **Robin boundary**: $\left\{\theta VC - \frac{\partial^{\alpha-2}}{\partial x^{\alpha-2}}\left[\theta D \frac{\partial(e^{\kappa x}C)}{\partial x}\right]\right\}\Big\|_{x=L} = g_1(t)$, defining the co-existence of an advective source (located outside of the upstream boundary and moving at a constant rate $V$) and an immobile diffusive source (located at the upstream boundary). | **Partially reflective boundary**: $\theta D \frac{\partial^{\alpha-1}(e^{-\kappa x}A)}{\partial(-x)^{\alpha-1}} e^{\kappa x}\Big\|_{x=L} = 0$, representing a partially free exit boundary. Diffusive particles cannot exit the boundary $x = L$, but are reflected near the boundary (to capture the diffusive source at the upstream boundary); advective particles, however, can exit the boundary $x = L$ freely, to capture the advective source outside $x = L$. |
| | **Infinite boundary**: $C\|_{x=-\infty} = 0$, with both advection and dispersion contribution to the mass flux in the domain ($L < x < R$) via the upstream boundary at $x = L$. | **Free boundary**: $A\|_{x=-\infty} = 0$, for infinite domains with advective & dispersive particles freely crossing the upstream boundary at $x = L$ (also called "a fully free exit boundary"). |
| Right (down-stream) | **Dirichlet boundary**: $C\|_{x=R} = g_2(t)$, representing a stagnant source reservoir or a mass sink term (with $g_2(t) = 0$, defining the absorption well or a groundwater barrier) at the downstream boundary. | **Absorbing boundary**: $A\|_{x=R} = 0$. A mass sink term in the forward model at the outlet transforms to a load term (with an initial probability of 1) in the backward model. |
| | **Neumann boundary**: $-\frac{\partial^{\alpha-2}}{\partial x^{\alpha-2}}\left[\theta D \frac{\partial(e^{\kappa x}C)}{\partial x}\right]\Big\|_{x=R} = g_2(t)$, representing diffusive flux leaving the system (with zero advective flux), which can define an impermeable layer at the outlet. | **Fully reflective boundary**: $\left[V\theta A - \theta D \frac{\partial^{\alpha-1}(e^{-\kappa x}A)}{\partial(-x)^{\alpha-1}} e^{\kappa x}\right]\Big\|_{x=R} = 0$, to completely close the outlet; so, no particles can exit the outlet from the internal domain and no external sources located downstream of the downstream boundary. |
| | **Robin boundary**: $\left\{\theta VC - \frac{\partial^{\alpha-2}}{\partial x^{\alpha-2}}\left[\theta D \frac{\partial(e^{\kappa x}C)}{\partial x}\right]\right\}\Big\|_{x=R} = g_2(t)$, representing both advective and diffusive flux leaving the system, due for example a pumping well. | **Partially reflective boundary**: $-\theta D \frac{\partial^{\alpha-1}(e^{-\kappa x}A)}{\partial(-x)^{\alpha-1}} e^{\kappa x}\Big\|_{x=R} = 0$. This partially reflective boundary is functionally analogous to the fully reflective boundary since the reversed flow direction, to remove any external pollutant sources. |
| | **Infinite boundary**: $C\|_{x=+\infty} = 0$, with both advection and dispersion contribution to the mass flux in the domain ($L < x < R$) via the downstream boundary at $x = R$, which is applicable for a site whose dimension is much longer than the pollutant displacement. | **Free boundary**: $A\|_{x=R} = 0$. This can be one of the predominant backward boundary conditions for real-world applications, where no physical boundaries exist or can be identified for forward pollutant transport with a limited scale in a regional-scale aquifer or river corridor. |

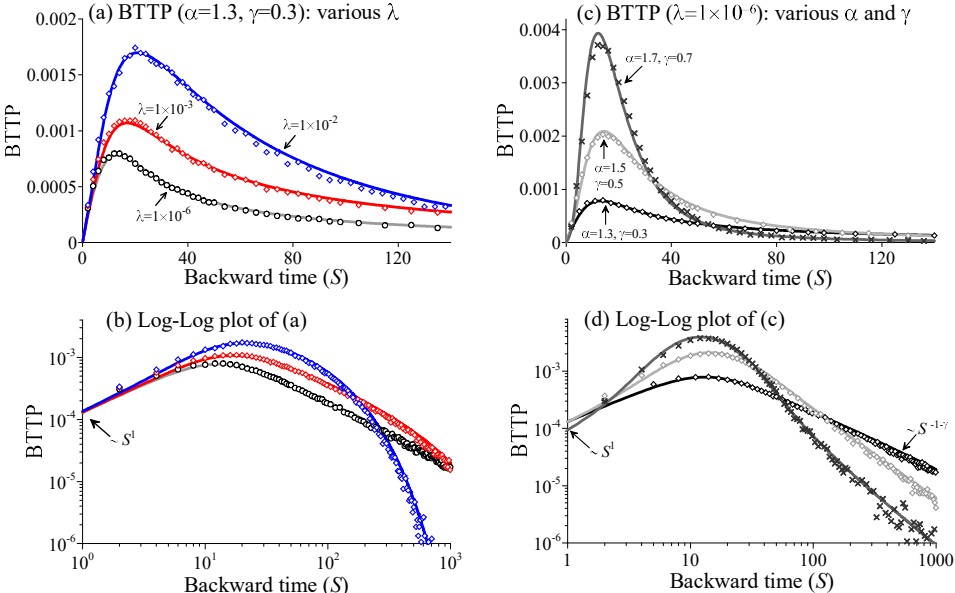


**Figure 1**. **Solver validation 1**: Lagrangian solutions (symbols) versus the Eulerian solutions (lines) for the 1-$d$ backward model (14a) with various truncation parameters $\lambda$ (a), and various subordination index $\alpha$ and time index $\gamma$ (c). The other model parameters that remain unchanged in these cases are as follows: velocity $V = 1$, scaling factor $\sigma^* = 1$, the spatial truncation parameter $\kappa = 1 \times 10^{-7}$, and the backward travel distance is $L = 10$. (b) and (d) are the log-log plot of (a) and (c),

respectively, to show the tailing. Free exit boundary conditions are used in these cases, and parameters are dimensionless here.

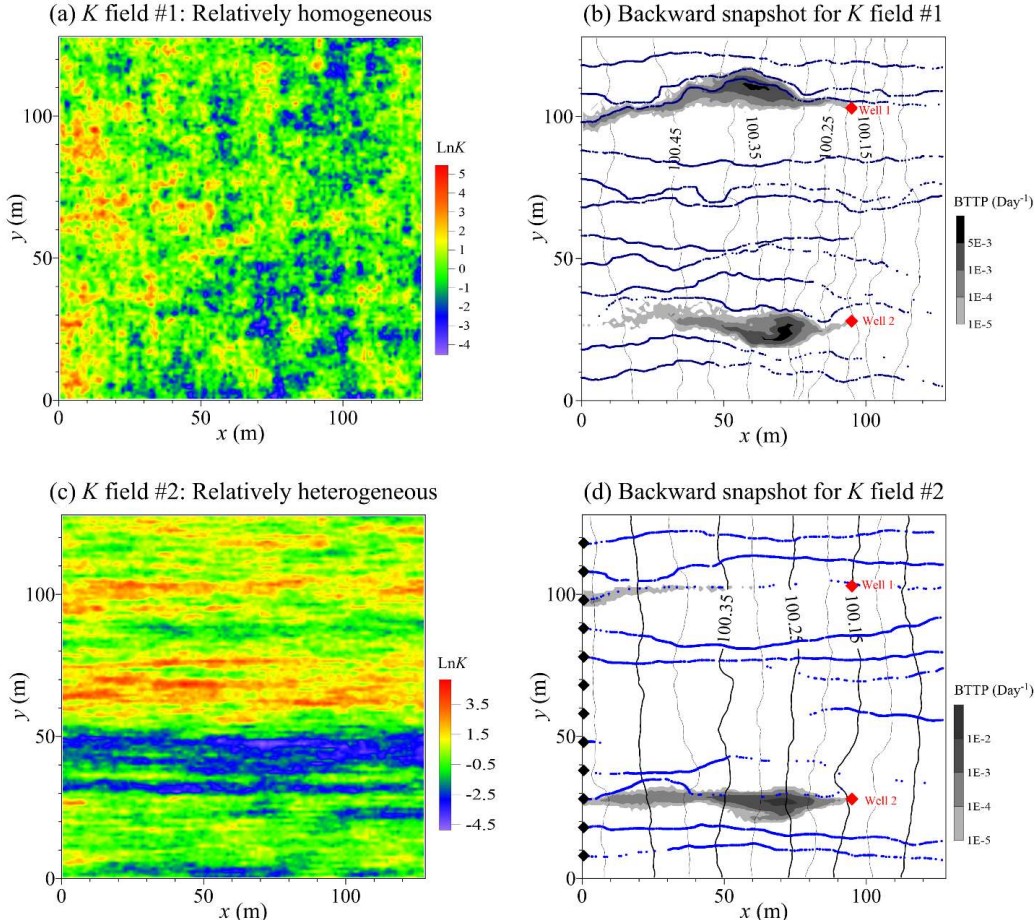

**Figure 2**. **Solver validation 2**: Two cases of operator-fractional Brownian fields (a) and (c). The corresponding backward particle tracking plume using the Lagrangian solvers for *K* field #1 and #2 is plotted in (b) and (d), respectively. In (b) and (d),

black lines represent the hydraulic head calculated by MODFLOW, blue dotted lines denote the streamlines) starting from the left boundary (shown by the black diamonds in (d)), and the red diamonds show the location of two monitoring wells.



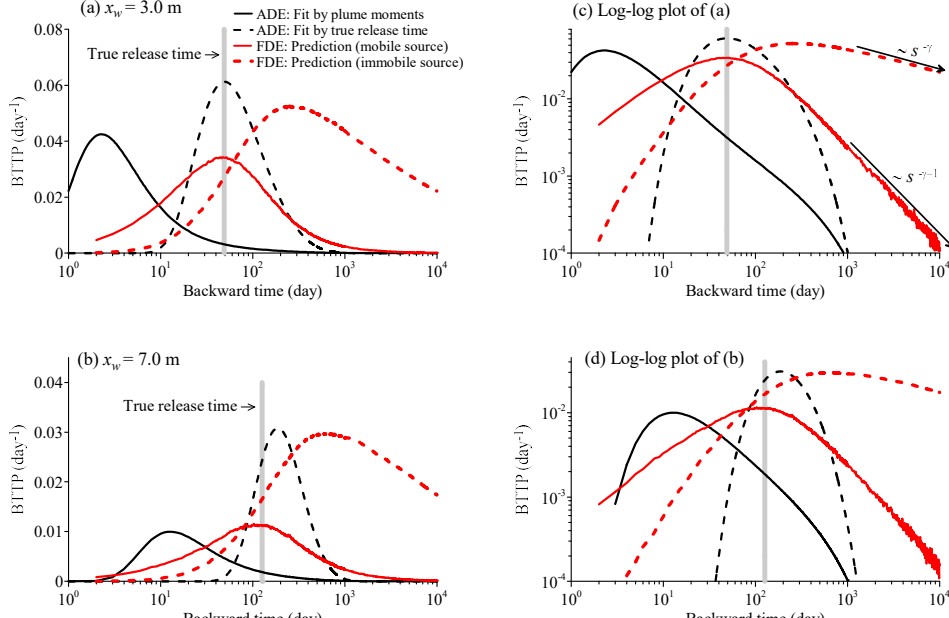

**Figure 3**. **BTTP Application 1:** MADE-1 aquifer: The calculated BTTP using the adjoint 1-d S-FDE (red lines) and the adjoint 1-d ADE (black line) for the observation well located at $x_w = 3.0$ m (a) and $x_w = 7.0$ m (b). (c) and (d) are the log-log plot of (a) and (b), respectively, to show the tailing behavior. The vertical grey bar denotes the true release time. The solid red line represents the BTTP for a mobile source, and the dashed red line represents the BTTP for an immobile source.






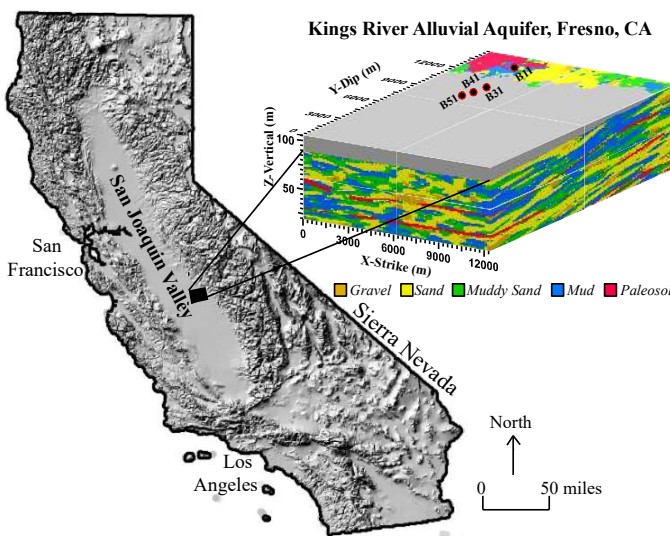

**Figure 4**. **BTTP Application 2:** KRAA - Location and the multiscale 3-*d* hydrofacies model for the Kings River alluvial aquifer, Fresno County, California.

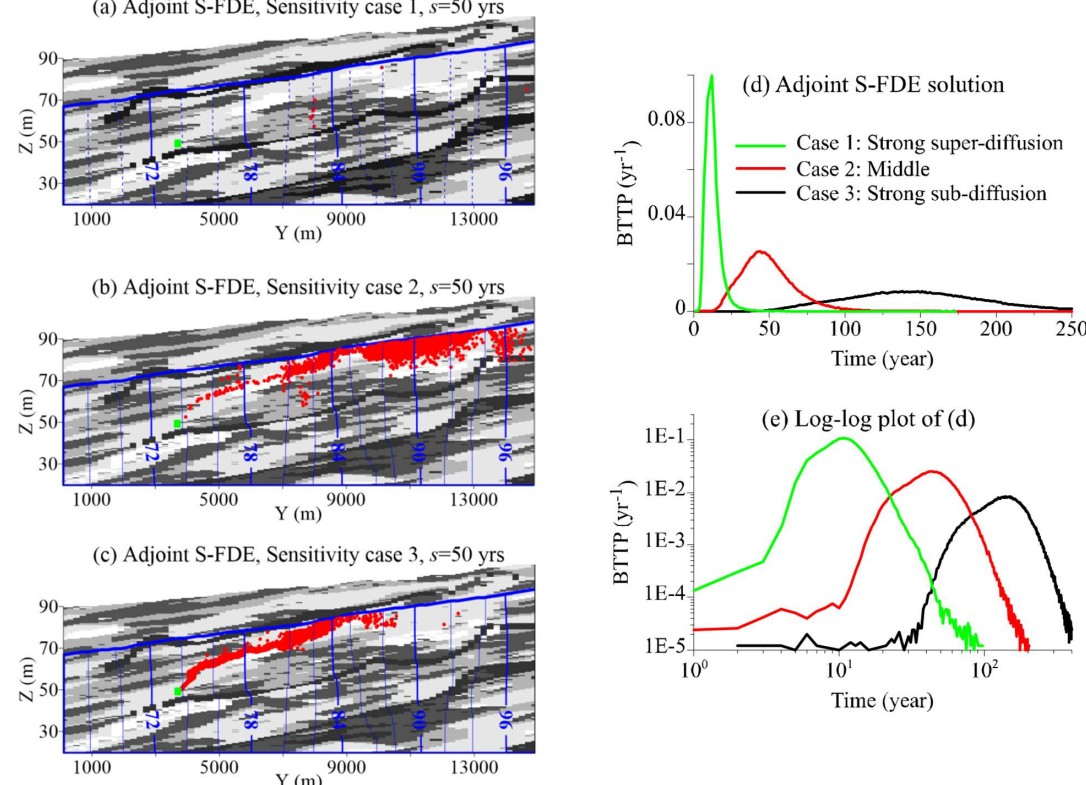

**Figure 5**. **BTTP Application 2:** KRF: Snapshot (project of particle plumes on the vertical cross section located at strike *X*=3,700 m shown in the hydrofacies model in **Figure 4** of backward particle tracking at the backward time *s*=50 yrs using the adjoint S-FDE (11a) for Case 1 (a), Case 2 (b), and Case 3 (c). The right plots show the snapshot of backward particle tracking at time *s*=50 yrs using the adjoint ADE with the dispersivity $\alpha_L = \alpha_T$ =0.4 m (d), 0.04 m (e), and 0.004 m (f). In all cases, 5,000 particles were released initially at *s*=0. The green rectangle in each plot represents the well screen (with a length of 0.5 m) where the groundwater sample is collected.



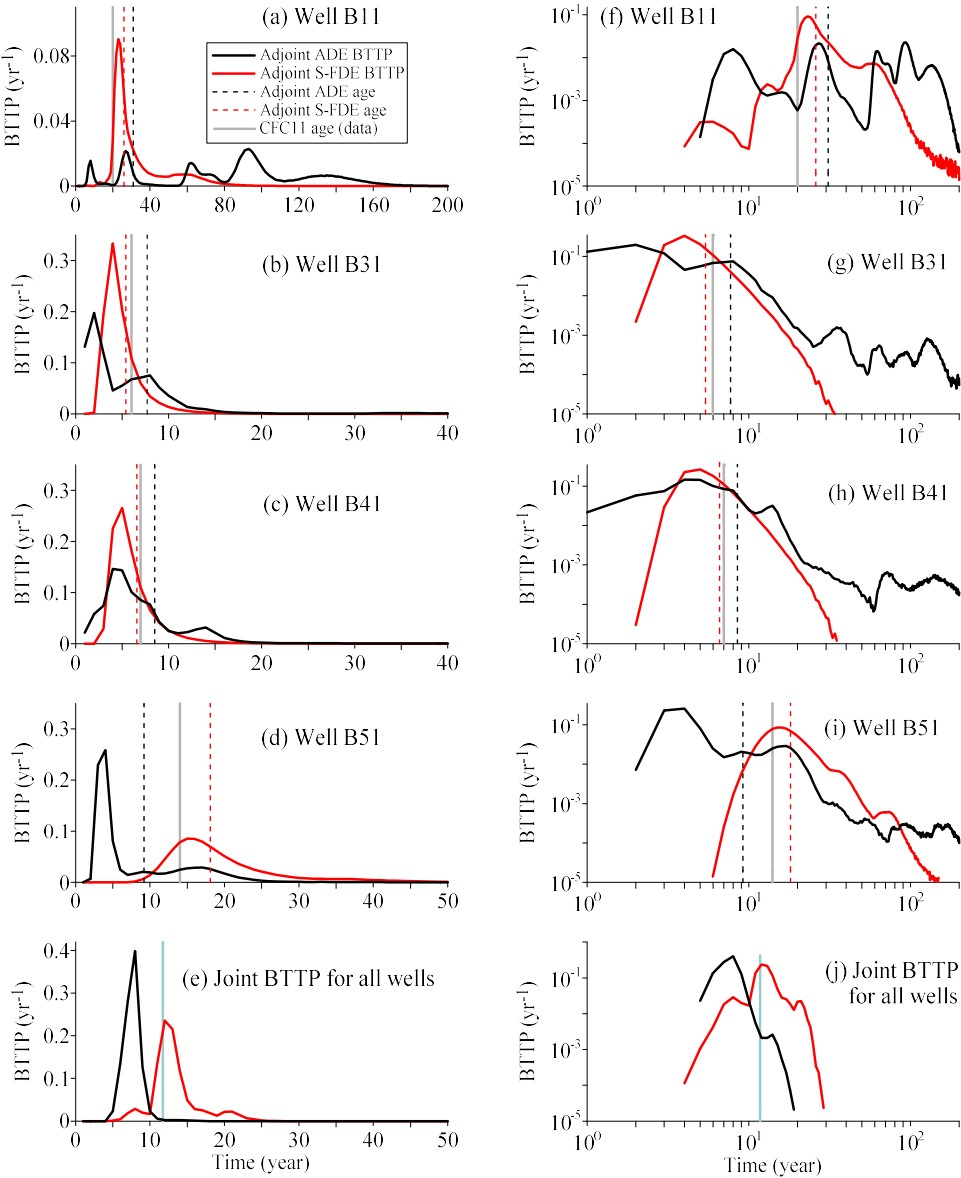

**Figure 6**. **BTTP Application 2:** KRF: the simulated BTTP using the adjoint S-FDE (red line) and the adjoint ADE (black line) for Well B11 (a), B31 (b), B41 (c), and 51 (d). The right plot is the log-log version of the left plot, to show the tailing. The vertical lines show the CFC-11 age measured in the lab (vertical grey line), estimated by the adjoint S-FDE (dashed red line), and estimated by the adjoint ADE (dashed black line).



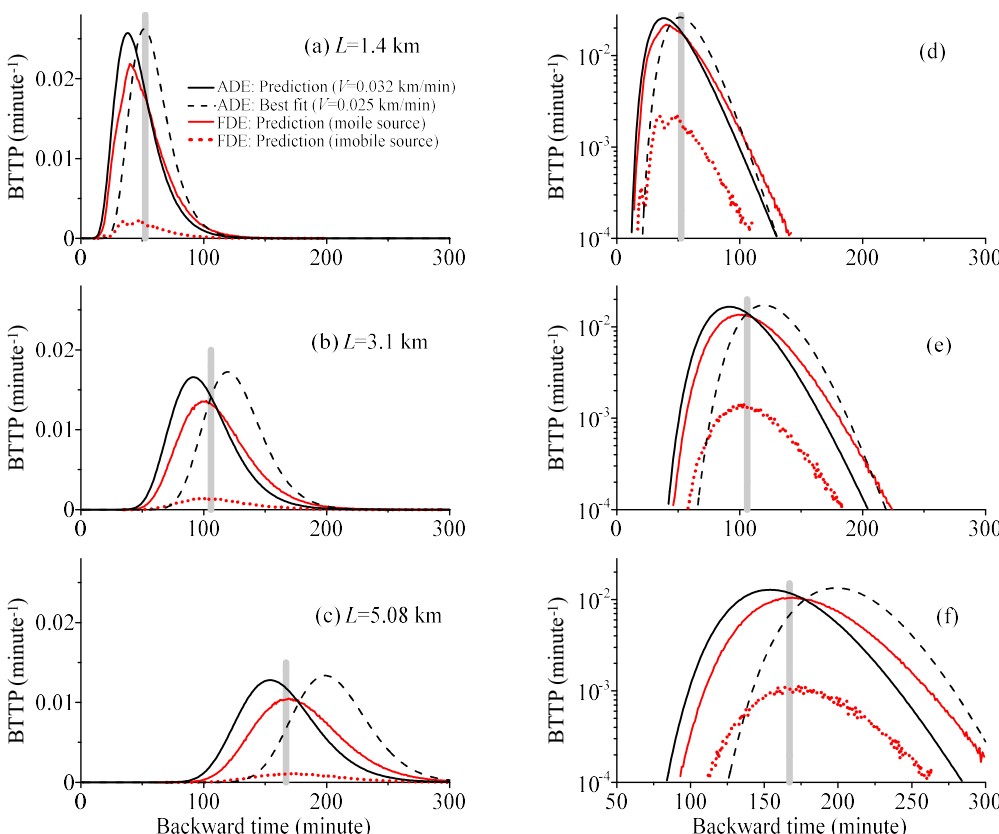

**Figure 7. BTTP Application 3** - Red Cedar River: the simulated BTTP using the adjoint S-FDE (red lines) and the adjoint
ADE (black lines) for the backward travel distance of *L*=1.4 km (a), 3.1 km (b), and 5.08 km (c). The right plot is the semi-
log version of the left plot, to show the tailing. The vertical bar in each plot shows the true release time. In the legend, "FDE:
Prediction (mobile source)" represents the predicted BTTP using the adjoint S-FDE for a mobile source, and "FDE: Prediction
(immobile source)" represents the predicted BTTP for an immobile source.






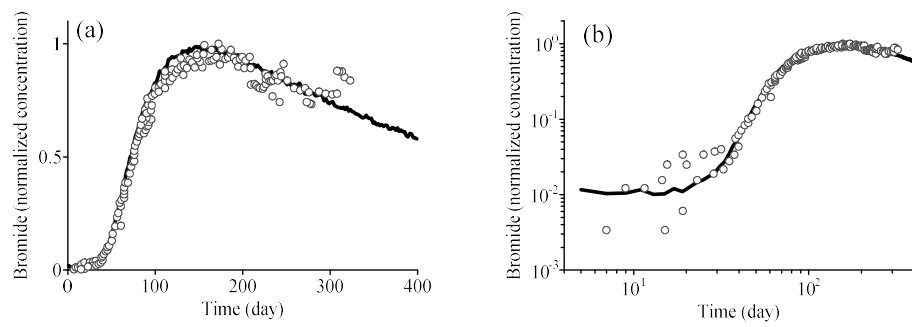

**Figure 8**. **BLP Application 1**: SHOAL test site: the measured (symbols) vs. the best-fit (line) bromide breakthrough curve using the vector model S-FDE (1a). (b) is the log-log plot of (a), to show the BTC tail.

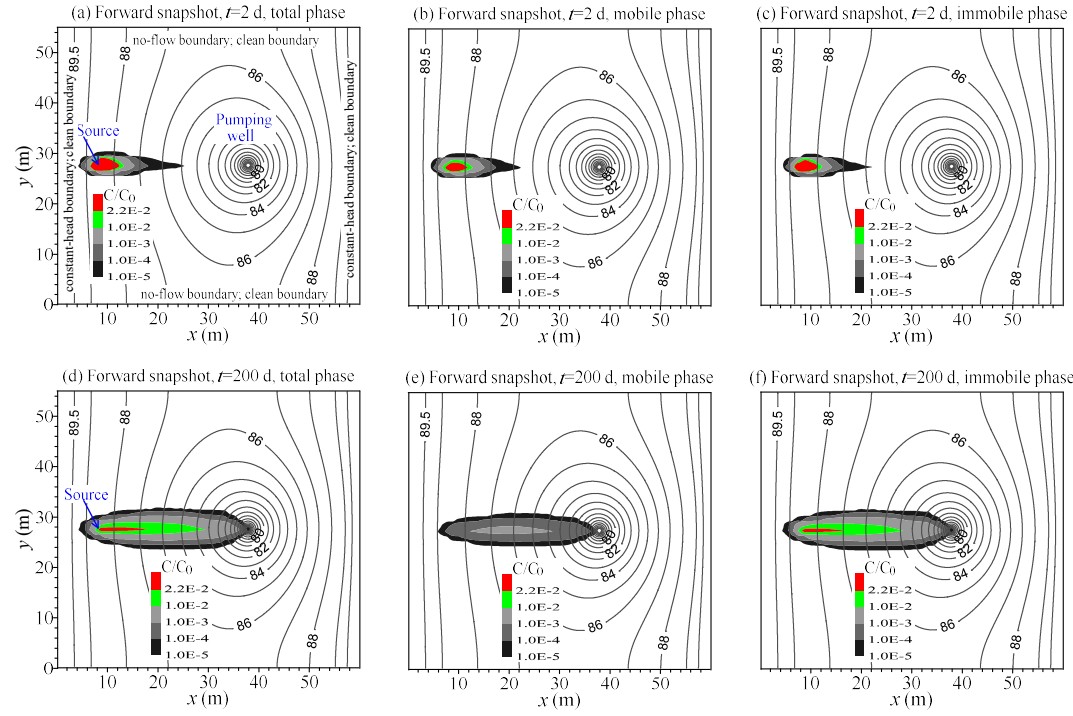


**Figure 9**. **BLP Application 1**: SHOAL test site: the modeled forward snapshot for the total phase (a), mobile phase (b), and immobile phase (c) at time $t$=2 days. (d), (e), and (f) show the snapshot at time $t$=200 days.

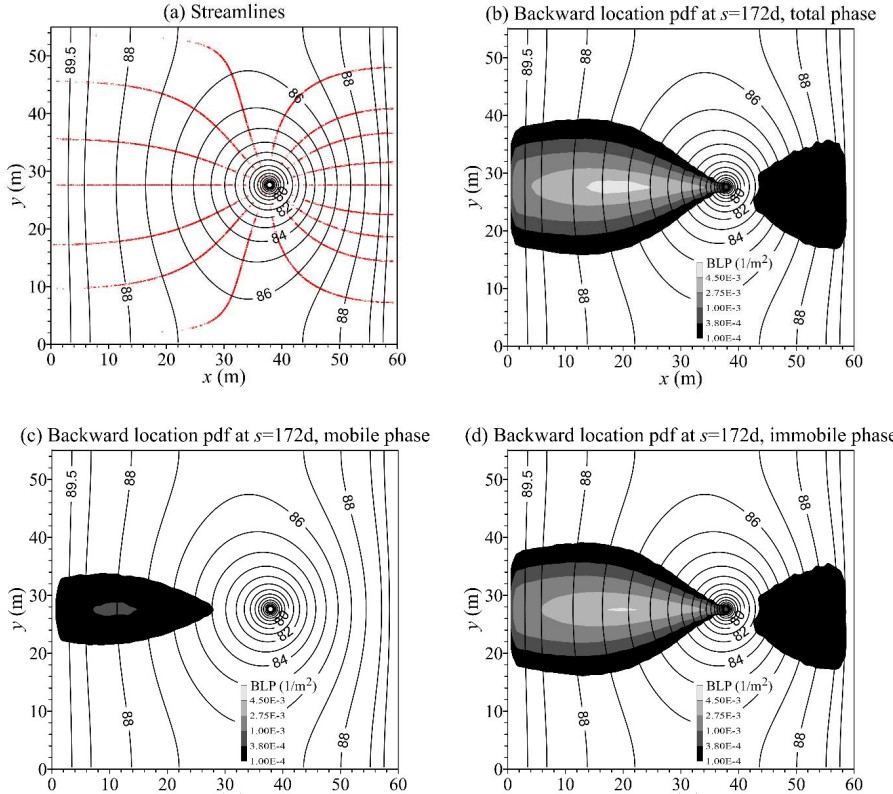

**Figure 10**. **BLP Application 1**: SHOAL test site: the modeled backward streamlines starting from the pumping well (a), and the calculated backward location probability density function (BLP) for pollutants located initially in the total phase (b), mobile phase (c), and immobile phase (d). It is noteworthy that there is a low concentration blob on the east side of the pumping well, due to the divergent flow in the backward model.



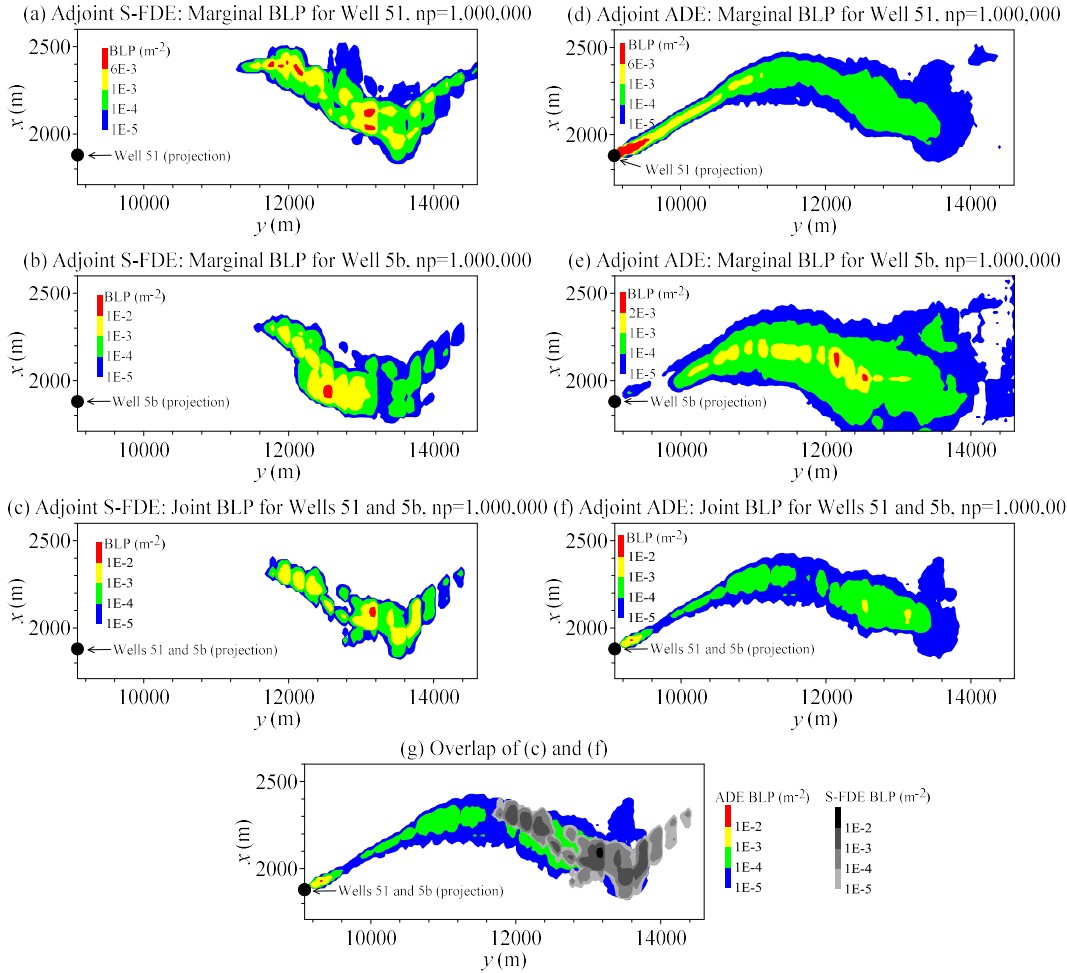

**Figure 11**. **BLP Application 2**: KRF: the simulated BLP using the adjoint S-FDE for Well B51 (a), B5b (b), and the adjoint BLP for Wells B51 and B5b (c). The adjoint ADE results are shown on the right plots. (g) is the overlap of plot (c) and (f). In the legend, "np" denotes the number of particles released in the Lagrangian solver.






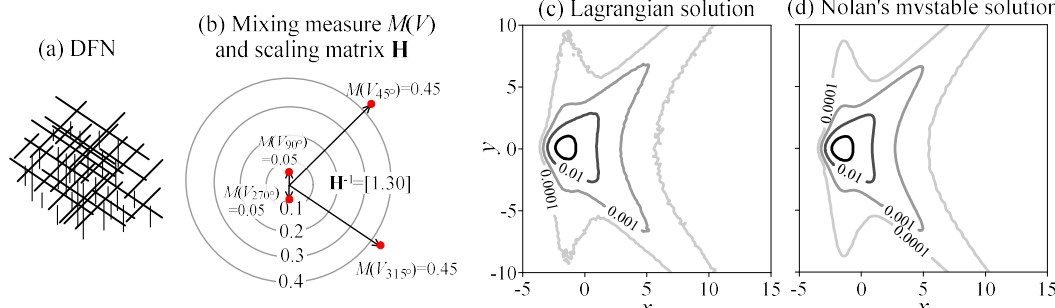

**Figure 12. Solver validation**: (a) shows the schematic diagram of a 2-*d* discrete fracture network. (b) is the polar plot of the discrete mixing measure and the scaling matrix. (c) is the Lagrangian solution of the multi-scaling S-FDE. (d) is Nolan's (1998) multivariate stable distribution.



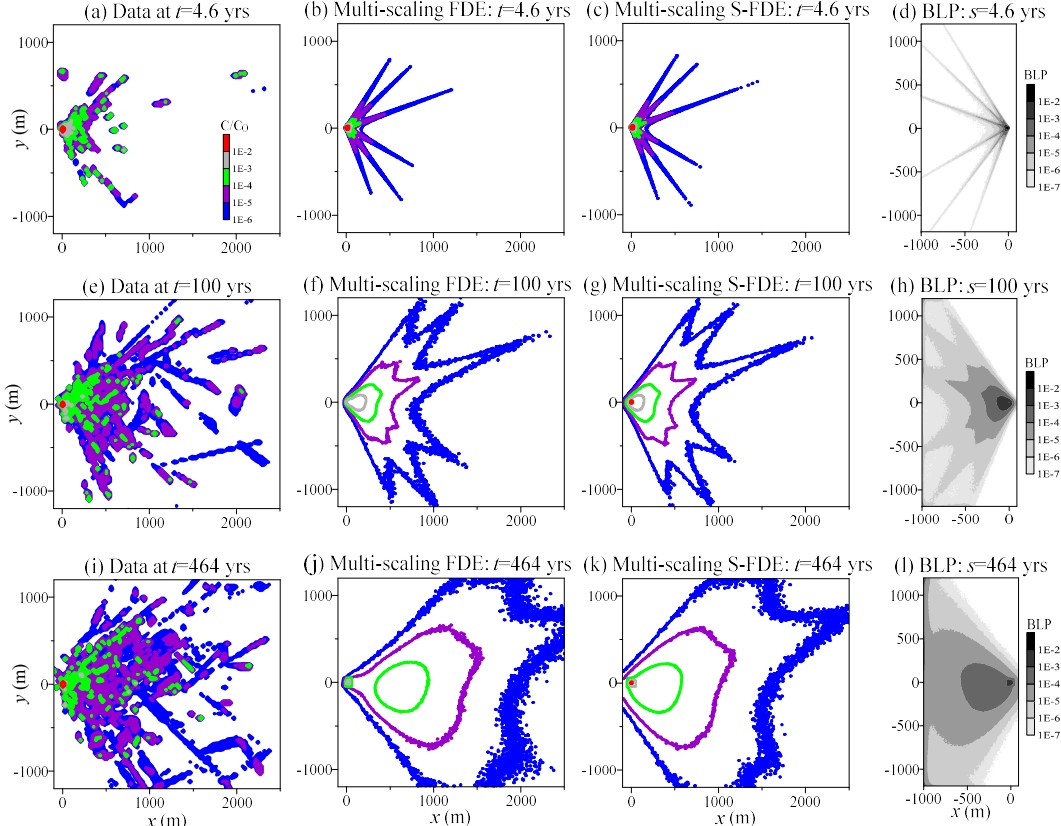

**Figure 13**. **Application of the multi-scaling S-FDE in DFNs**: (a) shows the average plume snapshot at time *t*=4.6 yrs from Monte Carlos simulations of pollutant transport in DFNs (Reeves et al., 2008). (b) and (c) are the best-fit solution using the multi-scaling FDE and multi-scaling S-FDE, respectively. (d) shows the resultant BLP using the multi-scaling S-FDE. The middle row (e)~(h) shows the result at a later time *t*=100 yrs, and the bottom row (i)~(l) shows the result at a later time *t*=464 yrs. Note that the model solutions in the middle and bottom rows are prediction results using parameters fitted in the top row.