# Peer review of "Adjoint subordination to calculate backward travel time probability of pollutants in water with various velocity resolutions"

_Hydrology and Earth System Sciences, 2023_

## Author Comment (AC1)

**RC1**: 'Comment on hess-2023-131', Anonymous Referee #1, 15 Sep 2023

Comments on the paper HESS-2023-131 entitled: "Adjoint subordination to calculate backward travel time probability of pollutants in water with various velocity resolutions"; by Y. Zhang et al.

The study proposes both a theoretical framework and applications to backtracking particles in a context of non-Fickian solute transport within diverse compartments of surface and subsurface water flows.

The concept of backtracking, mainly developed to retrieve transit time distribution of solutes reaching a given location, is not new. However, it is here developed in a context where sub- and super-diffusion could occur. A partially homogenized transport equation to mimic both sub- and super-diffusion could be that of an advection dispersion equation (ADE), complemented with fractional derivatives of the concentration with respect to both time and space coordinates. Sub-diffusion occurs mainly in systems where the solute is reversibly trapped by the porous medium, resulting in a squared displacement spread of solute proportional to time to power $\gamma$ (< 1). For its part, super-diffusion results from preferential high-velocity pathways with the consequence of a solute displacement spread to the power $\alpha$ (< 2) proportional to time.

The resulting equation simulating transport is a fractional derivative equation, subordinated to the flow velocity field, named as S-FDE, for which backtracking is theoretically grounded in an Adjoint (to concentration) state equation. The authors develop this adjoint S-FDE, which has some physical meaning if it is solved backward in time and over a reversed flow field. The development of the adjoint S-FDE is complemented by changes of boundary conditions compared to that of the forward problem. Those are documented by the authors for a simplified 1-D sweeping flow. Then, the authors propose diverse numerical test cases to solve the adjoint S-FDE in a Lagrangian framework moving particles over space within a reversed flow field and back in time.

First of all, I must acknowledge that the paper is very well crafted, not to say excellent. Two reasons for that.

- The mathematics are sound, clear and concise, even if a few shortcuts may persist. Nothing wrong in that because there is nothing that could not be retrieved by any attentive reader analyzing the paper in depth.

- The test cases are duly selected to show that an S-FDE and its adjoint companion, are what I could name a smart adaptive Physics. The fractional coefficients evolve according to the weakly versus highly resolution degree of the velocity field. Weakly-resolved fields tend to lower the fractional coefficients, when highly resolved fields render fractional coefficients close to 1 and 2, resulting in an "quasi" ADE mimicking solute transport. In short, the S-FDE and its adjoint report on an up-scaled Physics adapted to the prior knowledge we have on the system. The demonstration is clear in the paper and puts dots on the I and crosses on the T regarding the versatility of a S-FDE.

After perusing the manuscript twice (in truth, 2.5 times!), I only denoted a very few very minor points (minuscule points?) that could be easily cured within half an hour.

(1) I guess that $h$ in line 163, is some kind of objective function (as in an inverse problem). What is its form for an adjoint seeking the changes of the system if the source mass M0 is changed?

Reply: We appreciate the reviewer's valuable comments. We fully concur with the reviewer's assessment that the S-FDE and its adjoint model may upscale solute transport in saturated media characterized by varying degrees of heterogeneity and represented at various resolutions of flow velocity.

The symbol "$h$" denotes a functional, specifically a function of the system's state (i.e., function of several functions), denoted as $h(M_0, C)$ here. We have normalized the initial mass, $M_0$, for simplicity. Therefore, changes of $M_0$ do not affect the backward probability density functions (PDFs). In the adjoint equation, "$h$" plays a role in defining the detection location ($x'$) and time ($t'$) for pollutants within the backward model. For example, when computing the backward location PDF, "$h$" can be defined as follows (refer to Eq. (32) in Neupauer and Wilson [WRR, 1999]):

$$h = C^r(x, t)\, \delta(x - x')\, \delta(t - t'),$$

where $C^r$ represents the resident concentration, and $\delta$ is the Dirac delta function. Taking the Fréchet derivative of the equation above yields (shown in Appendix D in Neupauer and Wilson [1999]):

$$\frac{\partial h}{\partial C^r} = \delta(x - x')\, \delta(t - t'),$$

which defines the initial particle source in the Lagrangian solver for approximating the adjoint equation derived in this study.

(2) Table 1. First row. A typo I think. Change reference to Eq. 1a into 13a, and ref. to 11a into 14a.

Reply: Thanks for identifying this typo! In the revised manuscript, we updated (1a) to (13a) (now it becomes (6a)) and (11a) to (14a) (now it becomes (7a)) in the first row of Table 1.

(3) Lines 514-515. Probably a typo again mixing units in cm/s and m/s. Otherwise I do not understand how the value reported in line 514 would enter the range reported in line 515.

Reply: Thanks for identifying this typo. In the revised manuscript, we changed "cm/s" to "m/s" (line 578).

(4) 7, up left plot. Change "moile" in the posted caption by "mobile".

Reply: Thank you for catching that typo. In the revised manuscript, we've corrected "moile" to "mobile" in Figure 7 (line 1040).

(5) Line 495. In my opinion the notion of backward location probability (BLP) is not fully clear. As far I understand a BLP is simply a "standard" backtracking, then post-processed to get particle densities over elementary surfaces of volumes, then normalized so that the sum of these densities over the domain is one? If I am right, I suggest to mention it as such in the manuscript.

Reply: We concur with the reviewer that "BLP" represents a standard backtracking scheme, adhering to the established procedure for calculating particle number density-based PDFs in space. We have incorporated this statement into the revised manuscript (lines 555~556).

(6) Line 532. In essence, I do not see why the calculation of BLP for non-point pollutant differs from that of the test case before and held over a homogeneous radial flow field. The radial flow field backtracked from a pumping well should result in a single location of the source. The complex flow field of the KRAA, backtracked from a single monitored location should result in multiple probable sources. In both cases, the calculation via the adjoint S-FDE should not change. It is not what is suggested by the sentence in line 532.

Reply: We concur with the reviewer that the calculation using the adjoint S-FDE should remain consistent for non-point source pollutants. In Case 1, the forward model features a convergent flow field (due to pumping) which transforms into a divergent flow field in its backward counterpart. This divergence in flow can disperse particles to different locations, leading to multiple potential sources. We have included this clarification in the revised manuscript (lines 593~594).

---

## Author Comment (AC2)

**RC2**: 'Comment on hess-2023-131', Anonymous Referee #2, 18 Sep 2023

The manuscript develops an adjoint subordinated fractional-dispersion equation (S-FDE) in order to estimate release times and source locations of contaminants in aquifers and rivers. The author first (Section 2) present the three-dimensional forward S-FDE and then derive its adjoint following the approach of Neupauer and Wilson and using fractional-order integration by parts, and a fractional-order extension of the divergence theorem. Then, a Lagrangian backward solver is presented based on the developments of the lead authors. The solver is validated by comparison to finite difference solutions of the S-FDE. In Section 3, the developed backward tracking methodology is then applied to three field scenarios to estimate the release history of pollutants, and groundwater age. Section 4 discusses extensions of the proposed method to identify pollutant source locations, and multi-scale subordinated models, relevant for fractured media. This is an interesting contribution that adds to the literature on S-FDEs and source and release time identification in aquifers and rivers. In the following, I list a few comments and recommendations:

Comments:

(1) Line 103: Could the authors give a physical explanation of the meaning of the space-fractional advection term and the subordination to the velocity field? This is important because in field applications, solute transport is typically advection-dominated.

Reply: We thank the reviewer for offering helpful feedback that improved the presentation of this work. In the revised manuscript (lines 120-124), we added the following explanation regarding the subordination term in Eq. (1a), which can be directly equated to the space fractional derivative when the equation simplifies to one dimension:

"*It is worth noting that pollutant particles undergo advective displacement controlled by local mean velocity, with individual particles migrating along various flow paths in a heterogeneous medium, leading to random mechanical dispersion due to local speeds deviating from the mean velocity. Eq. (1a) assumes a (tempered) α-stable density distribution for random mechanical dispersive jumps, rescaled by the mean local velocity. This (tempered) α-stable density encompasses both Gaussian and power-law densities as two end members.*"

(2) Line 126 and following: The detailed derivations could be moved to an appendix.

Reply: Done. We moved the detailed deviations to Appendix A.

(3) Lines 123-124: It is not clear what the authors mean here. Molecular diffusion should model hydrodynamic dispersion? I assume the space fractional derivative should account for dispersion. This should be clarified.

Reply: We revised this statement (lines 267~268), and it now reads as follows:

"*Notably, if molecular diffusion is not negligible, it can be included in Eq. (1), combining with the subordination term responsible for mechanical dispersion to define hydrodynamic dispersion.*"

We concur with the reviewer that the space fractional derivative explains super-diffusion resulting from rapid displacement along preferential flow paths, which can typically overshadow molecular diffusion's influence on particle dispersion.

(4) Lines 146 and 152: When the authors refer to fractional-order integration and integration by parts of the spatial derivatives, do they mean the use of the divergence theorem and its fractional-order extensions? This should be clarified.

Reply: We used integration by parts, whether fractional or integer order, rather than the divergence theorem. For instance, Eq. (5) (now Eq. (A2)) employed fractional-order integration by parts, which doesn't involve vector field flux through a closed surface. Eq. (6) (now Eq. (A3)) applied integer-order integration by parts in the first equality and the Green's divergence theorem in the second. We clarified this in the revised manuscript (line 717).

(5) Section 2.2: This section refers extensively to previous work by the lead author. It would be instructive for the reader if the authors could provide the Lagrangian equations that are implemented in the solver.

Reply: Done. We added the temporal Langevin equation that describes the non-Markovin displacement of solute particles over time in the revised manuscript (lines 259~262).

(6) Lines 297 and 302: What is meant by relatively homogeneous/heterogeneous? How are the K-fields generated and how are they characterized (log-K variance, correlation length, etc.)?

Reply: We generated the K-fields using the 2D fractional Brownian motion (fBm) random field method developed by Zhang et al. (2019a). Particularly, log-normal random K values were distributed in space using the Fourier filter function. The Hurst parameter in the filter function defines the spatial correlation of K values: a relatively 'homogeneous' K-field exhibits weak correlation (e.g., Fig. 2a), while a 'heterogeneous' K-field displays strong correlation (e.g., Fig. 2c). We have included this clarification in the revised manuscript (lines 324~326).

(7) Line 300: Do the authors add diffusion to capture hydrodynamic dispersion? This needs to be clarified.

Reply: Yes, we have done this. It's necessary because molecular diffusion is a component of hydrodynamic dispersion. Please see also our response for Question 3.